# Nintedanib induces gene expression changes in the lung of induced-rheumatoid arthritis–associated interstitial lung disease mice

Shintaro Mikami[1], Yoko Miura[2], Shinji Kondo[3], Kosuke Sakai[1], Hiroaki Nishimura[1], Hiroyuki Kyoyama[1], Gaku Moriyama[1], Nobuyuki Koyama[1], Hideki Noguchi[3], Hirotsugu Ohkubo[4], Satoshi Kanazawa[2], Kazutsugu Uematsu[1]*

1 Department of Pulmonary Medicine, Saitama Medical Center, Saitama Medical University, Kawagoe, Saitama, Japan, 2 Department of Neurodevelopmental Disorder Genetics, Nagoya City University Graduate School of Medical Sciences, Nagoya, Aichi, Japan, 3 Center for Genome Informatics, Joint Support Center for Data Science Research, Research Organization of Information and Systems, Mishima, Shizuoka, Japan, 4 Department of Respiratory Medicine, Allergy and Clinical Immunology, Nagoya City University Graduate School of Medical Sciences, Nagoya, Aichi, Japan

* kuematsu@saitama-med.ac.jp

**Data Availability Statement:** The sequence datasets used in this study were deposited in DDBJ under the accession number DRA012991.

## Abstract

Nintedanib is a multi-tyrosine kinase inhibitor widely used to treat progressive fibrosing interstitial lung diseases because it slows the reduction in forced vital capacity. However, the prognosis for patients treated with nintedanib remains poor. To improve nintedanib treatment, we examined the effects of nintedanib on gene expression in the lungs of induced-rheumatoid arthritis–associated interstitial lung disease model mice, which develop rheumatoid arthritis and subsequent pulmonary fibrosis. Using next-generation sequencing, we identified 27 upregulated and 130 downregulated genes in the lungs of these mice after treatment with nintedanib. The differentially expressed genes included *mucin 5B* and heat shock protein 70 family genes, which are related to interstitial lung diseases, as well as genes associated with extracellular components, particularly the myocardial architecture, suggesting unanticipated effects of nintedanib. Of the genes upregulated in the nintedanib-treated lung, expression of *regulatory factor X2*, which is suspected to be involved in cilia movement, and *bone morphogenetic protein receptor type 2*, which is involved in the pathology of pulmonary hypertension, was detected by immunohistochemistry and RNA in situ hybridization in peripheral airway epithelium and alveolar cells. Thus, the present findings indicate a set of genes whose expression alteration potentially underlies the effects of nintedanib on pulmonary fibrosis. It is expected that these findings will contribute to the development of improved nintedanib strategies for the treatment of progressive fibrosing interstitial lung diseases.

## Introduction

Interstitial lung diseases are a large group of mostly progressive diseases of known or unknown causes that are characterized by chronic inflammation and fibrosis of the pulmonary

**Funding:** This work was supported by grants-in-aid from the Ministry of Education, Culture, Sports, Science and Technology (MEXT)/JSPS KAKENHI to Satoshi Kanazawa (Grant Numbers: JP26461470, 23591444, and 17K09982), and to Yoko Miura (17K16055), and a research grant from Nagoya City University (Grant Number 1943005) to Satoshi Kanazawa.

**Competing interests:** KU and HO received research funding and honoraria from Nippon Boehringer Ingelheim Co., Ltd., and SK received research funding from Boehringer Ingelheim Pharma GmbH & Co. KG. All other authors have nothing to disclose. This does not alter our adherence to PLOS ONE policies on sharing data and materials.

interstitium. Chronic interstitial lung diseases, including idiopathic interstitial pneumonias originating from an unknown etiology and collagen vascular diseases, mostly lead to irreversible fibrotic lesions and a very poor prognosis. Although the mean survival time is 3–5 years in patients with idiopathic pulmonary fibrosis, it varies greatly among patients, making accurate assessments of prognosis difficult [1]. Although attempts to identify the causes of interstitial lung disease have been made, the pathobiological mechanisms remain elusive, leading to a range of treatment approaches. Antifibrotic drugs, nintedanib and pirfenidone, are expected to improve prognosis, and evidence has accumulated regarding their treatment efficacy in patients with interstitial lung diseases; however, in clinical practice, their efficacies vary greatly and adverse events often inhibit their treatment [2, 3].

Nintedanib is a multi-tyrosine kinase inhibitor that targets platelet-derived growth factor receptor (PDGFR), vascular endothelial growth factor receptor (VEGFR), and fibroblast growth factor receptor (FGFR), and it is used clinically to slow the decrease of forced vital capacity in patients with idiopathic pulmonary fibrosis as well as in systemic scleroderma [4]. Similar effects have been reported in patients with progressive fibrosing interstitial disease of various origins [5]. Although the indications for nintedanib have been extended, its cost-effectiveness and frequent induction of adverse events hinder its clinical use. Thus, a better understanding of the mechanisms regulating the efficacy of and adverse events related to nintedanib treatment is needed. Anticipating that alteration of gene expression in the nintedanib-treated lung provides a clue to elucidate mechanisms regulating efficacy and adverse events of nintedanib treatment, we performed a comprehensive analysis of gene expression in the nintedanib-treated lung of a mouse model with interstitial pneumonia to identify predictors of treatment response or adverse events associated with nintedanib. Genome-wide association study is an important means of examining the relationship between gene expression and pathogenesis, and it has been used to elucidate the genetic mechanisms underlying many diseases, including idiopathic interstitial pneumonia [6]. Peyser et al. used single-cell sequencing and a bleomycin-induced pulmonary fibrosis model to characterize molecular response to fibrotic injury [7]. In that study, treatment of nintedanib reduced the bleomycin-induced cluster shift of fibroblasts to the extracellular matrix-enriched cluster, and no specific upregulation of gene expression was observed in pulmonary fibrosis lesion cells. A tissue-based sequencing approach has yet to be used to examine the gene expression alterations in nintedanib-treated lung.

Several mouse models, in which interstitial pneumonia was mostly induced by bleomycin administration via the respiratory tract, have already been used to validate the efficacy of nintedanib, including one with rheumatoid arthritis–associated lung fibrosis, in which nintedanib was found to reduce collagen levels in the lungs [8, 9]. The D1CC×D1BC mouse is a double-homozygous transgenic mouse carrying human class II major histocompatibility complex transactivator and the murine B7.1 gene under the control of the type II collagen promoter and enhancer [10, 11]. Immunization of these mice with bovine type II collagen induces rheumatoid arthritis and subsequent pulmonary fibrosis in this mouse model, thereby named the induced-rheumatoid arthritis–associated interstitial lung disease (iRA-ILD) model. Histopathological and biochemical analyses have shown that these mice develop nonspecific interstitial pneumonia pattern represented by lung inflammation [12].

Here, we examined gene expression in the nintedanib-treated lung of iRA-ILD mice. We identified 27 upregulated and 130 downregulated genes in the nintedanib-treated lung and conducted functional analyses of the identified genes to elucidate their potential roles in the activity of nintedanib and the induction of adverse events. We also examined the tissue expressions of several of the identified genes. The present findings are expected to be useful for the development of predictors of treatment response or adverse events associated with nintedanib.

## Materials and methods

### Mice and treatment schedule

All mouse experiments were performed according to the rules and regulations of the Fundamental Guidelines for Proper Conduct of Animal Experiments and Related Activities in Academic Research Institutions under the jurisdiction of the Ministry of Education, Culture, Sports, Science and Technology, Japan, and were approved by the Committee on the Ethics of Animal Experiments of Nagoya City University. D1CC×D1BC mice (8–12 weeks after birth and no distinction between male and female) were anesthetized with isoflurane and then immunized with 0.01 mg/mouse of bovine type II collagen (Collagen Research Center, Japan) and an equal dose of complete Freund's adjuvant (Becton, Dickinson and Company, NJ, USA) [10, 11]. The immunization time point was set at week 0, and bovine type II collagen booster injections containing incomplete Freund's adjuvant (Becton, Dickinson, and Company) were administered at weeks 3, 6, 9, and 12. The mice were then divided into two groups: nintedanib- and vehicle-treated. Nintedanib, supplied by Boehringer Ingelheim (Germany), was dissolved in 0.5% methylcellulose (Wako, Japan) and orally administered at a daily dose of 90 mg/kg from weeks 35 to 43. Blood was collected from the jugular vein for measurement of serum surfactant protein D (SP-D) levels (Yamasa, Japan). For assessment of fibrosis, the right lobe of the lung was used for the histopathological analysis, while the left lobe was used for gene expression analysis by RNA sequencing. Lungs were fixed in 4% paraformaldehyde, and 2-μm-thick paraffin sections were stained with hematoxylin and eosin and Masson's trichrome. Images showing the area of fibrosis represented in blue by Masson's trichrome staining were captured using a BZ-X analyzer (Keyence, Japan) and analyzed using ImageJ, Fiji. Data were calculated as blue ratio, with blue-stained area divided by total lung area [13]. All animal procedures, including those on D1CC×D1BC mice for in-house breeding, have been approved by the Laboratory Animal Facility of Nagoya City University.

### RNA sequencing and expression profiling

For RNA extraction, samples of around 2–3 mm$^3$ were cut from the periphery of the frozen lungs of 3 vehicle-treated, 5 nintedanib-treated, 1 non-treated mouse, and 4 age-matched D1CC×D1BC mice, and each tissue sample was numbered. Total RNA was isolated by using an RNeasy Plus Mini Kit (Qiagen, Germany), and the integrity and purity of the total RNA were assessed by using an Agilent 2100 Bioanalyzer (Agilent Technologies, CA, USA) and a Qubit Fluorometer (Thermo Fisher Scientific). A library for RNA sequencing was prepared by using a TruSeq Stranded mRNA Preparation Kit (Illumina, CA, USA). Single reads (151 bp in length) were generated by using a MiSeq sequencer (Illumina). The reads were aligned to the mouse genome (mm10) by using TopHat [14], and computation of expression levels of annotated genes (Ensembl version 84) [15] and identification of differentially expressed genes were performed by using Cuffdiff [16]. All computational work was performed by using the DNA Data Bank of Japan supercomputer system [17].

### Database analyses

Functional enrichment analysis was performed within the platform of the Database for Annotation, Visualization and Integrated Discovery (DAVID) by using categories related to biological processes, cellular components, and molecular functions [18, 19]. The default value of 0.1 was used as the significance threshold in the expression analysis systematic explorer protocol. Pathway analyses were also conducted using the Kyoto Encyclopedia of Genes and Genomes and WikiPathways (http://www.wikipathways.org/index.php/WikiPathways). Each pathway was assigned a $p$-value, and a value of $< 0.05$ was considered to be statistically significant.

## Quantitative reverse transcription and polymerase chain reaction (RT-qPCR)

Total RNA was extracted from the lungs as described above. cDNA was synthesized by using PrimeScript Master Mix (Perfect Real Time, Takara Bio, Japan) and amplified. Gene expression levels were compared by using an intercalator method with TB Green Premix Ex Taq II (Takara Bio) on an Applied Biosystems 7500 Fast Real-Time PCR System (Thermo Fisher Scientific). A calibration curve was constructed for comparative quantification, and *glyceraldehyde-3-phosphate dehydrogenase* (*Gapdh*) expression was used for normalization. Primer sets with the following sequences were purchased from Takara Bio (forward, reverse): *regulatory factor X2* (*Rfx2*), `GCCAGCATCACCAGCAGTACA` and `TCACAGTGCCTGCGATACACC`; *bone morphogenetic protein receptor type 2* (*Bmpr2*), `CCATCTGAGACCTTGCTATGCTGTA` and `AATGTCAGCGTTCATAGTGGCATC`; and *Gapdh*, `TGTGTCCGTCGTGGATCTGA` and `TTGCTGTTGAAGTCGCAGGAG`. RT-qPCR expression levels were compared by using Dunnett's multiple comparison test among the age-matched control, vehicle-treated, and nintedanib-treated groups. Statistical analysis was performed using the EZR software (Saitama Medical Center, Jichi Medical School, Japan), and the significance level was set at $p < 0.05$ [20].

## Western blotting

Lung samples were homogenized in RIPA buffer containing 20 mM Tris–HCl, pH 7.4, 150 mM NaCl, 1% Triton-X100, 0.5% deoxycholate sodium, 0.1% sodium dodecyl sulfate, 1 mM EDTA, 10 mM β-glycerophosphate, 10 mM NaF, 1 mM $Na_3VO_4$, and protease inhibitor cocktail. The extracts were sonicated for 10 min and centrifuged at $13,000 \times g$ for 15 min. Western blotting analysis was performed using an ECL system [13]. The following primary antibodies were used: mouse anti–BMPR2 (#GTX60415; GeneTex, CA, USA; 1:1000 dilution), rabbit anti-RFX2 (#12622-1-AP; Proteintech, IL, USA; 1:500 dilution), and mouse anti-α Tubulin (SC-8035; Santa Cruz Biotechnology, Tx, USA; 1:1000 dilution). Signals were detected using Immunostar Zeta (Fuji film, Japan) and an Amersham Imager 600 series imager (GE Healthcare, IL, USA). Statistical analyses of protein expression levels were performed using Fiji.

## Immunostaining

For multiplex staining of RFX2 in lung tissue, 2-μm-thick paraffin sections of lungs were reacted with the following primary antibodies: rabbit anti–E-cadherin (#3195, Cell Signaling Technology, MA, USA), rabbit anti–SP-C (#HP9050, Hycult Biotech, Netherlands), and rabbit anti-RFX2 (#12622-1-AP, Proteintech). For detection, Histofine Simple Stain Mouse MAX-PO secondary antibody (Nichirei, Japan) was used with an Opal multiplex fluorescent immunohistochemistry system (Akoya Biosciences, MA, USA) that included Opal 520, Opal 570, Opal 650, 1× amplification diluent, and AR6 buffer. For immunohistochemistry of BMPR2, de-paraffinized sections were stained with mouse anti-BMPR2 (#GTX60415, GeneTex), and the substrate solution was reacted with Histofine SAB-PO (M) Kit (Nichirei, Japan). For contrast staining, hematoxylin solution TypeM, (Mutoh Chemical Industry, Japan) was used. Images of stained tissues were recorded with a BZ-X710 fluorescence microscope.

## In situ hybridization

In situ hybridization for *secretoglobin family 1A member 1* (*Scgb1a1)* (#420351, ACD Bio, MI, USA), *surfactant protein C* (*Sftpc*) (#314101-C2), *Bmpr2* (#493061), and *Rfx2* (#1073021) transcripts was performed by using an RNAscope Multiplex Fluorescent Reagent Kit v2 (ACDBio).

## Results

### Effects of nintedanib on pulmonary fibrosis and gene expression

First, we examined the effects of nintedanib on pulmonary fibrosis and gene expression in iRA-ILD mice. Histological examination revealed that nintedanib treatment alleviated destruction of the alveolar architecture (Fig 1A: hematoxylin and eosin staining) and reduced the density of fibrotic regions (Fig 1B: Masson's trichrome staining), which is consistent with a previous report [12]. Next, using Masson staining blue ratio as an index of the level of fibrosis, we plotted the relationship between Masson stain blue ratio and serum SP-D level for 3 vehicle-treated (VA–VC) and 5 nintedanib-treated (NA–NE) iRA-ILD mice. This plot revealed a cluster of three of the nintedanib-treated mice (NA, NB, and NC) that was located far from the two vehicle-treated mice (VA and VB) (Fig 1C). Subsequent RNA sequencing revealed 16,263 genes (Ensembl version 84) that were expressed at ≥1 FKPM (fragments per kilobase of transcript per million of mapped reads) in 3 or more of the 26 datasets examined (2–4 replicates derived from 5 nintedanib- and 3 vehicle-treated individuals, and 1 non-treated individual) and confirmed the presence of two major clusters: one comprising mice NA, NB, and NC, and one comprising mice VA and VB (Fig 1D). Because mice ND, NE, and VC were not included in these two major clusters, we excluded those datasets from the following differential gene expression analysis.

A volcano plot was constructed to visualize the differential gene expression between the two clusters of mice. The results revealed significantly different gene expression for 157 genes, 27 of which were upregulated and 130 of which were downregulated (q-value < 0.05 by Cuffdiff) (Fig 1E and Table 1). Volcano plots were also constructed to visualize the differential gene expression between the nintedanib-treated mouse with the smallest Masson stain blue ratio and each of two vehicle-treated mice (NC vs. VB, Fig 1F; and NC vs. VA, Fig 1G). Among the three combinations of mice, 3 common upregulated genes (*Rfx2*, *Hspa1a*, and *Hspa1b*) and 13 common downregulated genes (*AC160982.1*, *Casq2*, *Fabp3-ps1*, *Hrc*, *Igkc*, *Igj*, *Myl3*, *Mb*, *Myl4*, *Myh6*, *Myom2*, *Tcap*, and *Tnnt2*) were observed (Fig 2A and 2B).

### Functional enrichment analysis and pathway analysis with DAVID

A functional enrichment analysis of 136 protein-coding genes from among the 157 up- or downregulated genes identified in the nintedanib-treated lungs was conducted within DAVID (Fig 3). The most significantly enriched terms were "system process" (47 genes) in the category "biological process", "contractile fiber part" (26 genes) in the category "cellular component", and "protein binding" (81 genes) in the category "molecular function". Pathway analysis using the Kyoto Encyclopedia of Genes and Genomes revealed that "cardiac muscle contraction" was the most significantly enriched pathway (10 genes) (Table 2). In another pathway analysis tool, WikiPathways, "calcium regulation in the cardiac cell" had the highest number of six genes (Table 2). This marked representation of cardiac functions suggested an unknown effect of nintedanib on the cardiovascular system. We therefore examined the expression of *Bmpr2*, which is reported to be associated with pulmonary hypertension [21] and is included among the 27 genes that were found to be significantly upregulated in the lung of nintedanib-treated iRA-ILD mice (Table 1A).

### Distribution of *Rfx2* and *Bmpr2* expression

Next, we examined the gene and protein expression of *Rfx2*, one of three common upregulated genes among the three combinations of nintedanib- and vehicle-treated mice based on progression of pulmonary fibrosis (*Rfx2*, *Hspa1a*, and *Hspa1b*; Fig 2A), and *Bmpr2*. RT-qPCR and

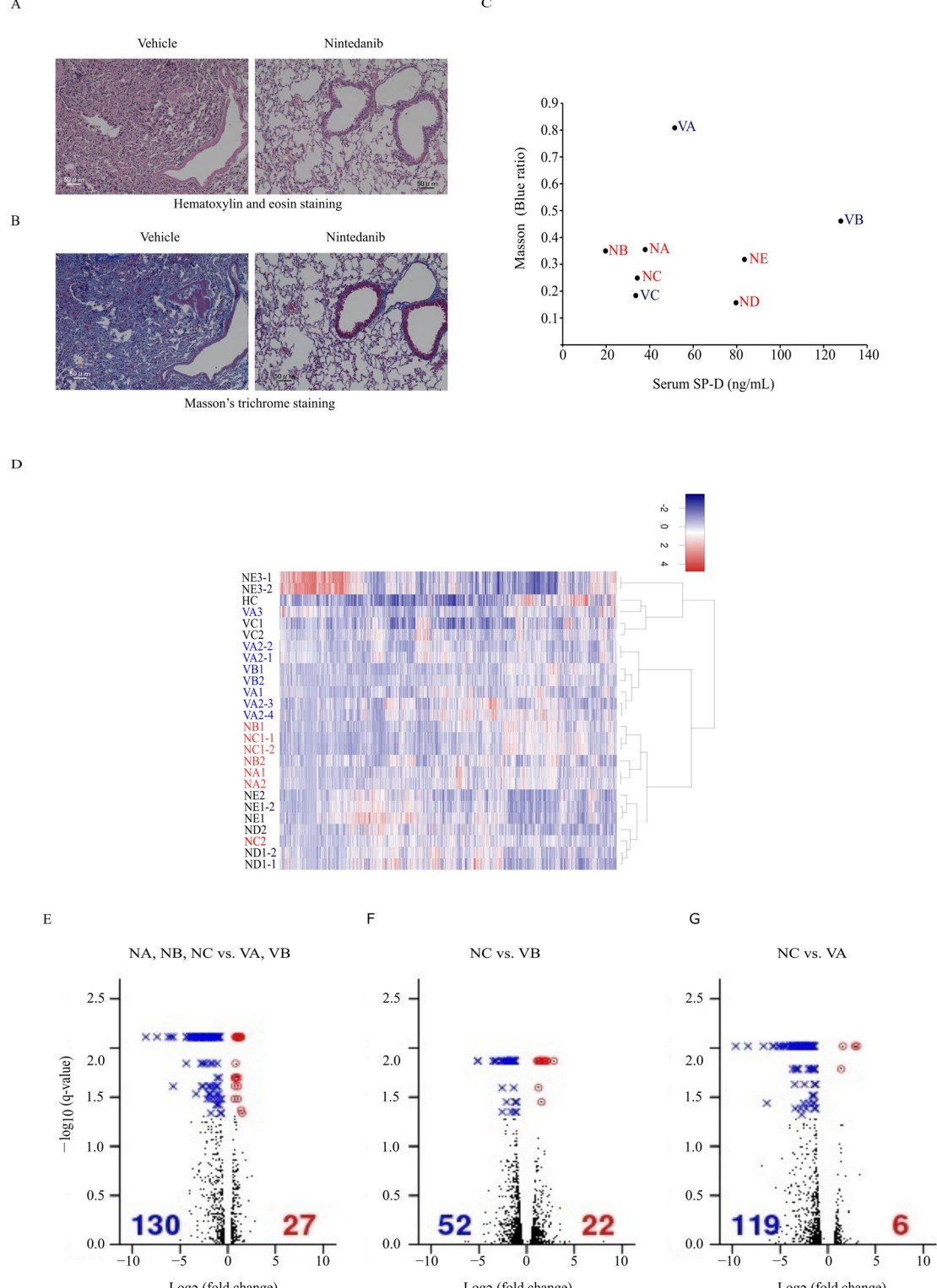

**Fig 1. Effects of nintedanib on gene expression in the lung of induced-rheumatoid arthritis–associated interstitial lung disease model mice.** (A, B) Histology of vehicle- and nintedanib-treated lung. (C) Relationship between surfactant protein D (SP-D) and Masson stain blue ratio as a measure of the progression of pulmonary fibrosis in nintedanib-treated (NA, NB, NC, ND, and NE) and vehicle-treated (VA, VB, and VC) mice. (D) Gene expression profile clustering for the 26 datasets derived from 5 nintedanib-treated, 3 vehicle-treated, and 1 non-treated (HC) mice. Two major clusters were confirmed: one comprising

nintedanib-treated mice, NA, NB, and NC, and another comprising vehicle-treated mice, VA and VB. Because mice ND, NE, and VC were not included in these clusters, they were excluded from the following analysis. (E–G) Volcano plots of genes differentially expressed between nintedanib- and vehicle-treated mice. Genes with $q$-value < 0.05 were considered significantly up- (red dots) or downregulated (blue crosses). (E) Comparison between the two major clusters observed among the nintedanib- and vehicle-treated mice. (F, G) Comparison between the mouse (NC) with the smallest Masson stain blue ratio as an index of progression of pulmonary fibrosis among the nintedanib-treated cluster and each of the vehicle-treated mice (VA and VB). The heatmap and volcano plots were drawn with the regHeatmap and plot programs in R, respectively (R Core Team. R: A language and environment for statistical computing. R Foundation for Statistical Computing, Vienna, Austria. 2014).

western blotting analyses showed that the expression of *Rfx2* was significantly downregulated in the vehicle-treated lung compared with that in the age-matched D1CC×D1BB lung and that this downregulation tended to be ameliorated by nintedanib treatment (Fig 4A–4C). To verify the location of cells with *Rfx2* expression, in situ hybridization was performed with *Sftpc* and *Scgb1a1* transcripts, which were used as an RNA-expression marker for type II alveolar cells and for airway secreting cells, respectively. In situ hybridization revealed that *Rfx2* was highly expressed in cells adjacent to *Scgb1a1*-positive cells localized in a basal cell–like position, and expression was observed in some type I and type II alveolar epithelial cells (Fig 4D). Subsequent immunostaining of lung tissue revealed that RFX2 was localized mainly in the bronchial epithelium, with some localization in the alveolar epithelium (Fig 4E). RT-qPCR and western blotting analyses also showed that *Bmpr2* expression tended to be downregulated in the vehicle-treated lung compared with that in the age-matched D1CC×D1BB lung, and that this downregulation also tended to be ameliorated by nintedanib treatment (Fig 5A–5C). *Bmpr2* was expressed mainly in type I alveolar epithelial cells, but some expression was also observed in type II alveolar epithelial and vascular endothelial cells, as shown by in situ hybridization (Fig 5D), and bronchial epithelium and alveolar cells, as shown by immunohistochemistry (Fig 5E).

## Discussion

Using RNA sequencing, we examined the expression of genes in the lungs of nintedanib-treated iRA-ILD mice and identified 27 upregulated and 130 downregulated genes. Using functional enrichment and pathway analyses, we identified the functions and pathways enriched for these differentially expressed genes. Transcripts of two upregulated genes of interest, *Rfx2* and *Bmpr2*, were found to be localized mainly in the bronchial epithelium and in peripheral type I alveolar cells, respectively. The RNA and protein expression levels of these genes were downregulated in iRA-ILD mouse lung compared with those in age-matched D1CC×D1BC mouse lung, and this downregulation tended to be ameliorated by nintedanib treatment.

Nintedanib is an inhibitor of multiple tyrosine kinases (i.e., PDGFR, VEGFR, and FGFR) and its effects on lung and lung fibroblasts have been validated by other groups by comparing the expressions of mesenchymal markers, cytokines, and chemokines, and by assessing changes in the expression levels of genes related to PDGFR, VEGFR, and FGFR [22]. Therefore, at the start of the present study we anticipated that we would observe changes in the expression of downstream genes associated mainly with PDGFR, VEGFR, and FGFR. However, functional enrichment and pathway analysis showed that fibrosis-related genes such as *Col1a1*, *Fn1*, and *Acta2* were not enriched in the terms "signaling pathway" and "focal adhesion," which are closely related to these tyrosine kinases, indicating that nintedanib also affects the expression of genes involved in other pathways not directly associated with these tyrosine kinases. The duration of nintedanib treatment in the present study may have blurred the direct effect on the target pathway by alternative compensation or acquisition of tolerance to

**Table 1. Genes with significantly different expression in the nintedanib-treated cluster (NA, NB, and NC) compared with that in the vehicle-treated cluster (VA and VB).**

**A, Upregulated genes**

| Gene | VA, VB FPKM | NA, NB, NC FPKM | Log$_2$ (fold change) | $q$-value |
|------|-------------|-----------------|----------------------|-----------|
| CXCR7 | 19.3151 | 33.6332 | 0.800158 | 0.00772067 |
| CXCR2 | 18.2111 | 33.9321 | 0.897829 | 0.00772067 |
| EGR1 | 70.688 | 133.5 | 0.917303 | 0.00772067 |
| LYVE1 | 63.8211 | 128.547 | 1.0102 | 0.00772067 |
| ZBTB16 | 1.43302 | 2.95718 | 1.04516 | 0.00772067 |
| IL1B | 31.2319 | 65.5559 | 1.06971 | 0.00772067 |
| RFX2 | 11.8997 | 25.6924 | 1.11041 | 0.00772067 |
| HSPA1B | 2.45952 | 5.58818 | 1.184 | 0.00772067 |
| WFDC17 | 115.445 | 267.074 | 1.21004 | 0.00772067 |
| GPR182 | 10.9326 | 26.776 | 1.29231 | 0.00772067 |
| OLFM2 | 6.69372 | 17.313 | 1.37098 | 0.00772067 |
| SLURP1 | 31.4862 | 82.9056 | 1.39675 | 0.00772067 |
| CEBPD | 45.0448 | 79.6934 | 0.823101 | 0.0143384 |
| BMPR2 | 16.2579 | 27.6254 | 0.764857 | 0.0200737 |
| GM12715 | 24.529 | 41.93 | 0.773497 | 0.0200737 |
| MSLN | 35.8908 | 65.1855 | 0.860937 | 0.0200737 |
| LRRN4 | 8.08446 | 14.8763 | 0.879788 | 0.0200737 |
| CD33 | 3.90997 | 8.31561 | 1.08867 | 0.0200737 |
| GM6091 | 0 | 9.50101 | infinity | 0.0200737 |
| UPK3B | 55.6659 | 94.6203 | 0.765355 | 0.0245176 |
| HSPA1A | 2.12857 | 4.4776 | 1.07284 | 0.0245176 |
| THBS1 | 18.9632 | 31.0342 | 0.710657 | 0.0329979 |
| BNC1 | 1.44643 | 3.0144 | 1.05937 | 0.0329979 |
| DOC2B | 0.354321 | 0.937176 | 1.40326 | 0.042824 |
| HAS1 | 1.03694 | 2.93801 | 1.50251 | 0.046029 |
| GM11966 | 0 | 60.9628 | infinity | 0.046029 |
| GM9836 | 0 | 1.13249 | infinity | 0.046029 |

**B, Downregulated genes**

| Gene | VA, VB FPKM | NA, NB, NC FPKM | Log$_2$ (fold change) | $q$-value |
|------|-------------|-----------------|----------------------|-----------|
| BPIFA1 | 1245.85 | 3.26086 | −8.57766 | 0.00772067 |
| SCGB3A1 | 2932.79 | 17.298 | −7.40553 | 0.00772067 |
| CFD | 91.4231 | 1.22822 | −6.21792 | 0.00772067 |
| CAR3 | 55.3028 | 0.891895 | −5.95434 | 0.00772067 |
| BPIFB1 | 84.6211 | 1.60889 | −5.71688 | 0.00772067 |
| HRC | 1.50571 | 0.0755911 | −4.31608 | 0.00772067 |
| ATP6V1B1 | 3.31325 | 0.172118 | −4.26677 | 0.00772067 |
| SLN | 138.152 | 8.84086 | −3.96593 | 0.00772067 |
| FABP4 | 74.8584 | 4.97829 | −3.91044 | 0.00772067 |
| LTF | 27.6337 | 1.86417 | −3.88982 | 0.00772067 |
| DHRS7C | 5.73937 | 0.415391 | −3.78835 | 0.00772067 |
| EEF1A2 | 7.31426 | 0.539155 | −3.76194 | 0.00772067 |
| MB | 104.893 | 8.10969 | −3.69313 | 0.00772067 |
| IGKV5-39 | 100.39 | 7.95728 | −3.65719 | 0.00772067 |
| IGKV1-88 | 123.208 | 10.6068 | −3.53803 | 0.00772067 |
| CKM | 16.6249 | 1.44354 | −3.52567 | 0.00772067 |
| HFE2 | 4.33859 | 0.381436 | −3.50771 | 0.00772067 |

(*Continued*)

**Table 1.** (Continued)

| | | | | |
|---|---|---|---|---|
| MYOM2 | 3.70856 | 0.356718 | −3.378 | 0.00772067 |
| TNNT2 | 15.1432 | 1.5366 | −3.30086 | 0.00772067 |
| SCN4B | 1.21398 | 0.126167 | −3.26634 | 0.00772067 |
| MYPN | 0.674664 | 0.0726609 | −3.21492 | 0.00772067 |
| CASQ2 | 14.6393 | 1.57942 | −3.21238 | 0.00772067 |
| SULT1D1 | 38.6526 | 4.20636 | −3.19992 | 0.00772067 |
| TNNI3 | 80.0224 | 8.80993 | −3.1832 | 0.00772067 |
| KCNJ3 | 0.867501 | 0.096145 | −3.17358 | 0.00772067 |
| PGAM2 | 14.2136 | 1.58361 | −3.16598 | 0.00772067 |
| MYLK4 | 2.39294 | 0.272587 | −3.134 | 0.00772067 |
| MYL3 | 42.6881 | 4.8646 | −3.13344 | 0.00772067 |
| LMOD2 | 3.11109 | 0.354867 | −3.13207 | 0.00772067 |
| TCAP | 31.6657 | 3.62429 | −3.12715 | 0.00772067 |
| TBX20 | 1.10388 | 0.127134 | −3.11816 | 0.00772067 |
| MYOZ2 | 21.7692 | 2.52089 | −3.11029 | 0.00772067 |
| HAMP2 | 14.9551 | 1.74189 | −3.10191 | 0.00772067 |
| MAPK10 | 2.19094 | 0.263876 | −3.05362 | 0.00772067 |
| KCNJ5 | 2.48496 | 0.300424 | −3.04815 | 0.00772067 |
| MYL4 | 271.307 | 32.8382 | −3.04648 | 0.00772067 |
| MYBPHL | 25.5662 | 3.12984 | −3.03008 | 0.00772067 |
| OBSCN | 2.30883 | 0.283226 | −3.02713 | 0.00772067 |
| CSRP3 | 43.5866 | 5.47211 | −2.99372 | 0.00772067 |
| MYBPC3 | 3.13885 | 0.394985 | −2.99036 | 0.00772067 |
| IGKV8-21 | 64.8776 | 8.33437 | −2.96058 | 0.00772067 |
| MYL7 | 178.84 | 23.3513 | −2.93709 | 0.00772067 |
| MYH6 | 29.0568 | 3.83181 | −2.92278 | 0.00772067 |
| CIDEA | 4.79633 | 0.637523 | −2.91138 | 0.00772067 |
| TRIM72 | 1.87988 | 0.250196 | −2.90951 | 0.00772067 |
| SRL | 10.1022 | 1.38155 | −2.87031 | 0.00772067 |
| SLC17A7 | 1.55908 | 0.213957 | −2.8653 | 0.00772067 |
| LDB3 | 11.904 | 1.66065 | −2.84162 | 0.00772067 |
| APOBEC2 | 3.67632 | 0.536149 | −2.77756 | 0.00772067 |
| ACTN2 | 6.56642 | 0.978013 | −2.74718 | 0.00772067 |
| H19 | 11.7258 | 1.75799 | −2.73769 | 0.00772067 |
| FNDC5 | 4.28137 | 0.736442 | −2.53943 | 0.00772067 |
| IGJ | 164.271 | 29.7197 | −2.46658 | 0.00772067 |
| AC160982.1 | 606.034 | 114.295 | −2.40663 | 0.00772067 |
| AC155333.4 | 49.2416 | 9.4281 | −2.38484 | 0.00772067 |
| TMC5 | 1.31172 | 0.252444 | −2.37743 | 0.00772067 |
| AC122260.1 | 160.319 | 31.3856 | −2.35277 | 0.00772067 |
| HPCAL4 | 3.76695 | 0.775159 | −2.28083 | 0.00772067 |
| FABP3-PS1 | 94.2604 | 19.6168 | −2.26456 | 0.00772067 |
| CMYA5 | 0.918198 | 0.193743 | −2.24466 | 0.00772067 |
| MUC5B | 2.94309 | 0.645161 | −2.1896 | 0.00772067 |
| GRIP2 | 1.11467 | 0.247066 | −2.17365 | 0.00772067 |
| THRSP | 10.3025 | 2.30766 | −2.15849 | 0.00772067 |
| ITGB1BP2 | 6.13918 | 1.3837 | −2.14951 | 0.00772067 |
| COX7A1 | 53.1048 | 12.5873 | −2.07687 | 0.00772067 |

(*Continued*)

**Table 1.** (Continued)

| | | | | |
|---|---|---|---|---|
| AC140374.1 | 138.69 | 33.0677 | −2.06837 | 0.00772067 |
| OIT1 | 3.62555 | 0.912052 | −1.99101 | 0.00772067 |
| RYR2 | 1.4109 | 0.358858 | −1.97513 | 0.00772067 |
| COX8B | 38.9729 | 10.6796 | −1.86761 | 0.00772067 |
| SCARA5 | 2.39048 | 0.709311 | −1.75281 | 0.00772067 |
| MYLK3 | 2.88567 | 0.856285 | −1.75275 | 0.00772067 |
| IGKC | 5168.26 | 1560.37 | −1.72779 | 0.00772067 |
| HSPB7 | 14.261 | 4.35892 | −1.71003 | 0.00772067 |
| CYP2E1 | 18.1495 | 5.5502 | −1.70932 | 0.00772067 |
| ANKRD1 | 33.0626 | 10.5035 | −1.65433 | 0.00772067 |
| RBM24 | 2.02133 | 0.694055 | −1.54218 | 0.00772067 |
| FSD2 | 1.96342 | 0.683569 | −1.52221 | 0.00772067 |
| LYPD2 | 142.763 | 51.703 | −1.46531 | 0.00772067 |
| CHRM2 | 3.70048 | 1.43571 | −1.36595 | 0.00772067 |
| IGHG2B | 22.6271 | 8.9014 | −1.34595 | 0.00772067 |
| LUM | 13.8944 | 5.54847 | −1.32434 | 0.00772067 |
| ACTC1 | 133.066 | 54.2248 | −1.29512 | 0.00772067 |
| CIDEC | 15.2732 | 6.66569 | −1.19618 | 0.00772067 |
| RETNLA | 198.488 | 90.59 | −1.13163 | 0.00772067 |
| PDK4 | 10.3249 | 4.71416 | −1.13105 | 0.00772067 |
| COX6A2 | 51.8812 | 24.7128 | −1.06995 | 0.00772067 |
| ATP2A2 | 95.2461 | 45.7116 | −1.0591 | 0.00772067 |
| IGHM | 385.226 | 191.572 | −1.00782 | 0.00772067 |
| FABP1 | 37.7763 | 18.7869 | −1.00776 | 0.00772067 |
| IFI27L2A | 1250.84 | 648.296 | −0.948171 | 0.00772067 |
| 2010107E04RIK | 556.286 | 318.941 | −0.802537 | 0.00772067 |
| KRT5 | 0.541073 | 0 | -infinity | 0.00772067 |
| RETN | 9.39548 | 0.462495 | −4.34446 | 0.0143384 |
| KBTBD10 | 1.85295 | 0.269106 | −2.78358 | 0.0143384 |
| 2310042D19RIK | 1.10467 | 0.181547 | −2.6052 | 0.0143384 |
| GM1078 | 2.02305 | 0.403901 | −2.32446 | 0.0143384 |
| IGLV1 | 98.2544 | 27.6526 | −1.82911 | 0.0143384 |
| SLC26A10 | 5.68793 | 1.92171 | −1.56552 | 0.0143384 |
| ASB2 | 6.83883 | 3.17949 | −1.10496 | 0.0143384 |
| CD300LG | 5.39903 | 2.51853 | −1.10012 | 0.0200737 |
| CNN1 | 35.3961 | 18.9262 | −0.903201 | 0.0200737 |
| A2M | 3.08174 | 0.0583869 | −5.72196 | 0.0245176 |
| TXLNB | 1.59361 | 0.241102 | −2.72458 | 0.0245176 |
| BVES | 1.90096 | 0.316698 | −2.58554 | 0.0245176 |
| CORIN | 1.02572 | 0.25851 | −1.98835 | 0.0245176 |
| NRAP | 1.23726 | 0.341576 | −1.85687 | 0.0245176 |
| AC174597.1 | 75.1934 | 25.2305 | −1.57543 | 0.0245176 |
| PPP1R1B | 3.56163 | 1.49967 | −1.2479 | 0.0245176 |
| ADCYAP1R1 | 3.67297 | 1.6976 | −1.11345 | 0.0245176 |
| NEK2 | 3.46607 | 1.61758 | −1.09947 | 0.0245176 |
| TNNI3K | 1.24423 | 0.124461 | −3.32149 | 0.0293047 |
| AC153855.1 | 42.1131 | 10.7539 | −1.9694 | 0.0293047 |
| AGR2 | 5.72943 | 1.51661 | −1.91754 | 0.0293047 |

(Continued)

**Table 1.** (Continued)

| | | | | |
|---|---|---|---|---|
| SLC25A34 | 0.607796 | 0.173024 | −1.81261 | 0.0293047 |
| BC048546 | 2.58243 | 1.1177 | −1.2082 | 0.0293047 |
| ENO3 | 23.9436 | 12.5342 | −0.93377 | 0.0293047 |
| SYNPO2L | 0.877739 | 0.20124 | −2.12488 | 0.0329979 |
| PRDM8 | 1.13032 | 0.326858 | −1.78999 | 0.0329979 |
| ADH7 | 4.02313 | 1.56129 | −1.36558 | 0.0329979 |
| NTRK2 | 5.1035 | 2.47072 | −1.04656 | 0.0329979 |
| PIGR | 15.7217 | 9.36853 | −0.746866 | 0.0329979 |
| AC113287.1 | 159.375 | 100.305 | −0.668039 | 0.0329979 |
| IGKV12-38 | 1.09571 | 0 | -infinity | 0.0329979 |
| GDPD2 | 9.96178 | 4.43505 | −1.16745 | 0.0377225 |
| ART3 | 42.6334 | 19.5891 | −1.12193 | 0.0377225 |
| FLNC | 1.98444 | 1.07624 | −0.882733 | 0.0377225 |
| SLC2A4 | 7.45498 | 2.18924 | −1.76777 | 0.046029 |
| CD34 | 79.9189 | 47.9767 | −0.736204 | 0.046029 |
| MT-RNR1 | 847.951 | 512.502 | −0.726424 | 0.046029 |
| ITIH4 | 22.3465 | 13.8304 | −0.692204 | 0.046029 |

FPKM, fragments per kilobase of transcript per million of mapped reads.

inhibitors. Although comprehensive gene expression analyses have been performed in fibrotic lesions and normal regions in lungs and in lung cell lines, at the present no reports exist discussing a comprehensive analysis of gene expression in nintedanib-treated lungs [23–25].

Although several of the downregulated genes detected in the present study are associated with the myocardial architecture, these genes are not reported to be directly involved in the pathogenesis of pulmonary fibrosis. Although the INPULSIS trial showed a low risk of adverse cardiovascular events, it has been noted that the risk of myocardial infarction tends to increase with subsequent accumulation of nintedanib-treated patients [26, 27]. On the other hand,

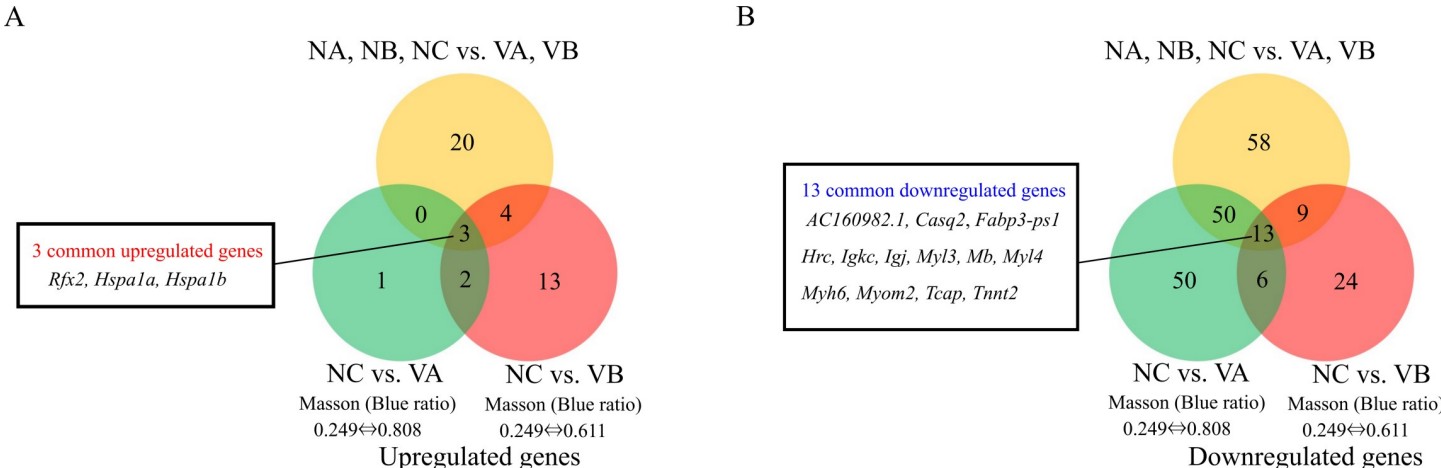

**Fig 2.** Venn diagrams showing overlap of genes up- (A) and downregulated (B) by nintedanib treatment between three combinations of nintedanib- (NA, NB, and NC) and vehicle-treated mice (VA and VB). Yellow, comparison between the 3 nintedanib-treated and 2 vehicle-treated mice comprising the two major clusters observed in the gene expression analysis. Green and red, comparison between the nintedanib-treated mouse with the smallest Masson stain blue ratio as an index of progression of pulmonary fibrosis and each of the vehicle-treated mice. Among the combinations, 3 common upregulated genes and 13 common downregulated genes were observed.

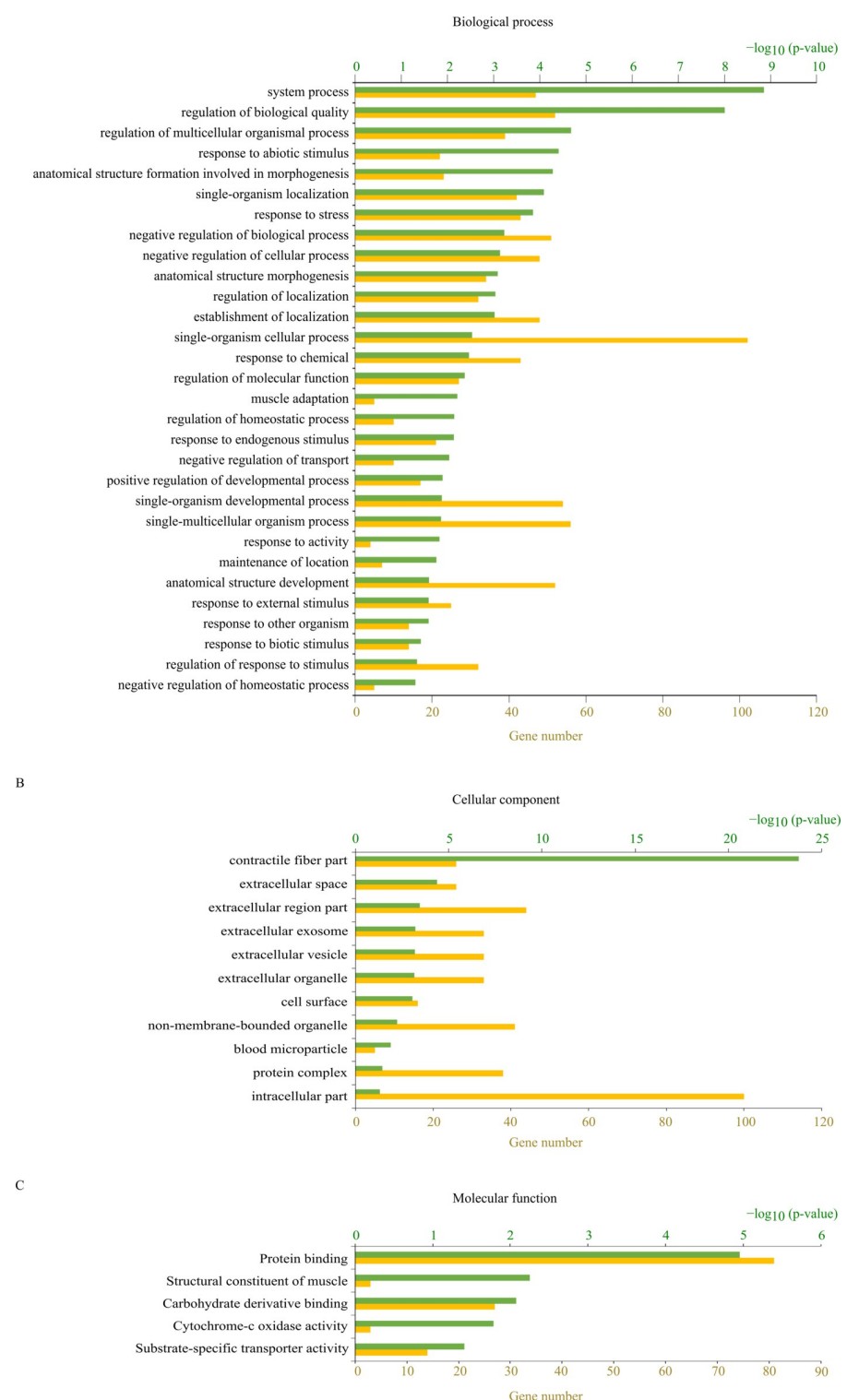

**Fig 3. Results of a functional enrichment analysis of genes differentially expressed between nintedanib- and vehicle-treated mice conducted within the Database for Annotation, Visualization, and Integrated Discovery (DAVID).** Functional enrichment was examined within the categories of (A) biological process, (B) cellular component, and (C) molecular function; *p*-values less than 0.05 were considered significant, and the number of genes annotated in each functional category is given.

**Table 2. Results of pathway analyses using Kyoto Encyclopedia of Genes and Genomes pathway or WikiPathways.**

| Kyoto Encyclopedia of Genes and Genomes Pathway | | |
| --- | --- | --- |
| Term | No. of genes | *p*-value |
| Cardiac muscle contraction | 10 | 1.63E-09 |
| Adrenergic signaling in cardiomyocytes | 8 | 4.21E-05 |
| Hypertrophic cardiomyopathy (HCM) | 6 | 1.79E-04 |
| Dilated cardiomyopathy | 6 | 2.26E-04 |
| Focal adhesion | 7 | 0.002564022 |
| Non-alcoholic fatty liver disease (NAFLD) | 6 | 0.003989727 |
| Legionellosis | 4 | 0.006596853 |
| cAMP signaling pathway | 6 | 0.010270069 |
| Oxytocin signaling pathway | 5 | 0.018236826 |
| MAPK signaling pathway | 6 | 0.026558059 |
| Estrogen signaling pathway | 4 | 0.028271323 |
| Circadian entrainment | 4 | 0.028271323 |
| Alzheimer's disease | 5 | 0.031162944 |
| Retrograde endocannabinoid signaling | 4 | 0.032112484 |
| Calcium signaling pathway | 5 | 0.032863034 |
| WikiPathways | | |
| Glycolysis | 4 | 1.29E-05 |
| Calcium regulation in the cardiac cell | 6 | 3.14E-05 |
| Adipogenesis genes | 5 | 2.29E-04 |
| Fatty acid omega oxidation | 2 | 3.46E-04 |
| MAPK signaling pathway | 5 | 6.50E-04 |
| Glycolysis and gluconeogenesis | 3 | 0.00108568 |
| Iron homeostasis | 2 | 0.001472475 |
| Apoptosis modulation by HSP70 | 2 | 0.002449189 |
| Fatty acid beta oxidation (streamlined) | 2 | 0.007647089 |
| IL-1 signaling pathway | 2 | 0.011212501 |
| Heart development | 2 | 0.016011843 |
| Non-odorant GPCRs | 4 | 0.024312584 |
| SIDS susceptibility pathways | 2 | 0.026115967 |
| Complement and coagulation cascades | 2 | 0.026115967 |
| Lung fibrosis | 2 | 0.026915045 |
| Insulin signaling | 3 | 0.027511103 |
| Hfe effect on hepcidin production | 1 | 0.028360134 |
| Alzheimer's disease | 2 | 0.038162217 |
| Ethanol metabolism resulting in production of ROS by CYP2E1 | 1 | 0.040268466 |
| Nicotine activity on dopaminergic neurons | 1 | 0.040268466 |
| PPAR signaling pathway | 2 | 0.043857694 |

reports of improvements in myocardial fibrosis with nintedanib are of clinical interest [28]. The results of a recent transcriptome analysis of the mouse pulmonary myocardium with respect to atrial fibrillation have much in common with the genes downregulated in the present study [29], and the effect of nintedanib on intrapulmonary blood vessels is noteworthy. A previous study reported the effect of nintedanib on pulmonary artery smooth muscle in a rat model of pulmonary arterial hypertension [30]. Although the mechanism of how nintedanib influence genes related to cardiovascular system is not clear at the present time, the results of

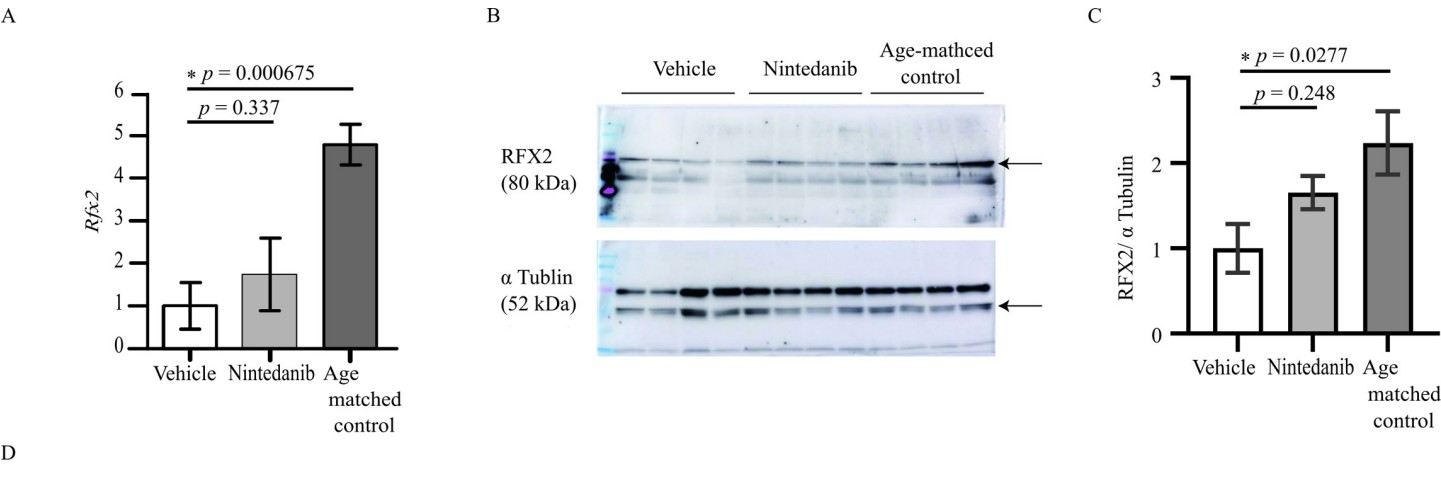

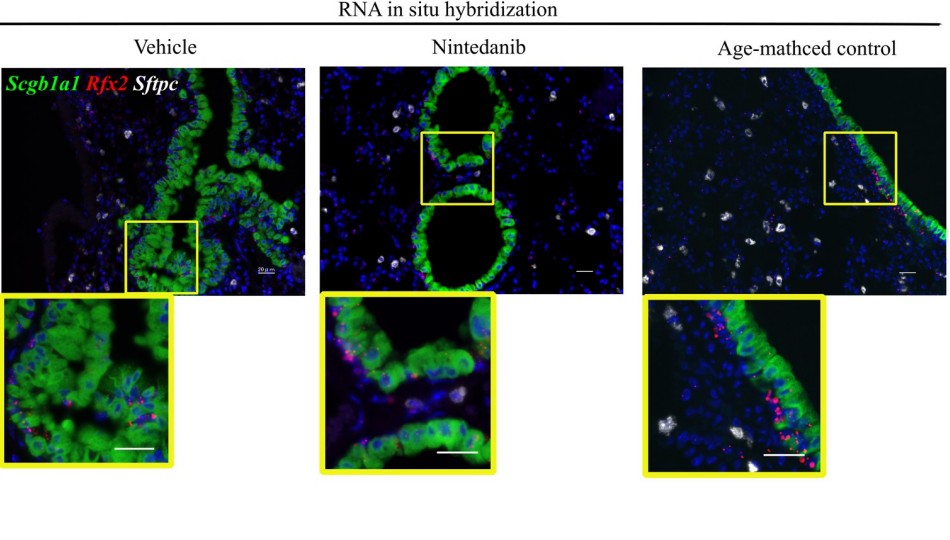

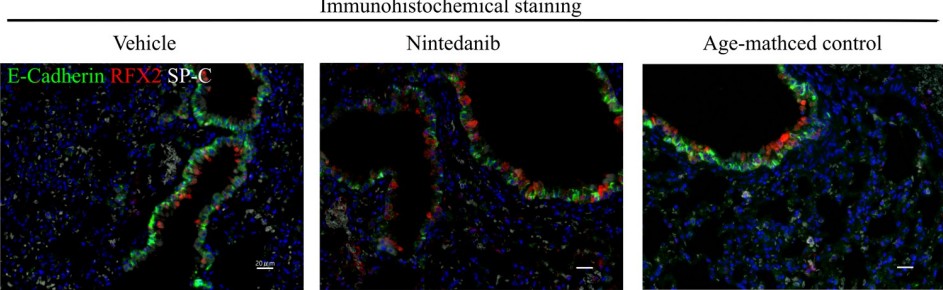

**Fig 4. *Rfx2* expression in lung.** (A–C) *Rfx2* expression in vehicle-treated, nintedanib-treated, and age-matched lung tissue. (A) Quantitative reverse transcription polymerase chain reaction analysis showed that *Rfx2* expression was significantly downregulated in vehicle-treated mouse lung (n = 3) compared with that in age-matched D1CC×D1BB mouse lung (n = 3), and that this downregulation tended to be attenuated by nintedanib treatment (n = 3). The vehicle- and nintedanib-treated mouse lungs were obtained from the mice used in the analysis shown in Fig 1C. (B, C) Western blot analysis confirmed the findings shown in (A): age-matched D1CC×D1BB mouse lung, n = 4; vehicle-treated mouse lung, n = 4; nintedanib-treated mouse lung, n = 4 Relative signal intensity was determined using α Tubulin as the loading control. Data are presented as mean ± standard error. *, $p < 0.05$ by Dunnett's test. (D) RNA in situ hybridization for *Rfx2* revealed that *Rfx2* was expressed strongly and weakly in cells adjacent to *Scgb1a1*-positive cells and in some type I and type II alveolar epithelial cells, respectively. *Sftpc* and *Scgb1a1* transcripts were used as an RNA-expression marker for type II alveolar cells and airway secreting cells, respectively. Scale bars = 20 μm. (E) Immunohistochemical staining revealed that RFX2 (red) was localized mainly in bronchial epithelium and somewhat in alveolar epithelium. The expressions of E-cadherin (green) and SP-C (white) were used as a marker for epithelial cells, and epithelial cells and type II alveolar epithelial cells, respectively. Scale bars = 20 μm.

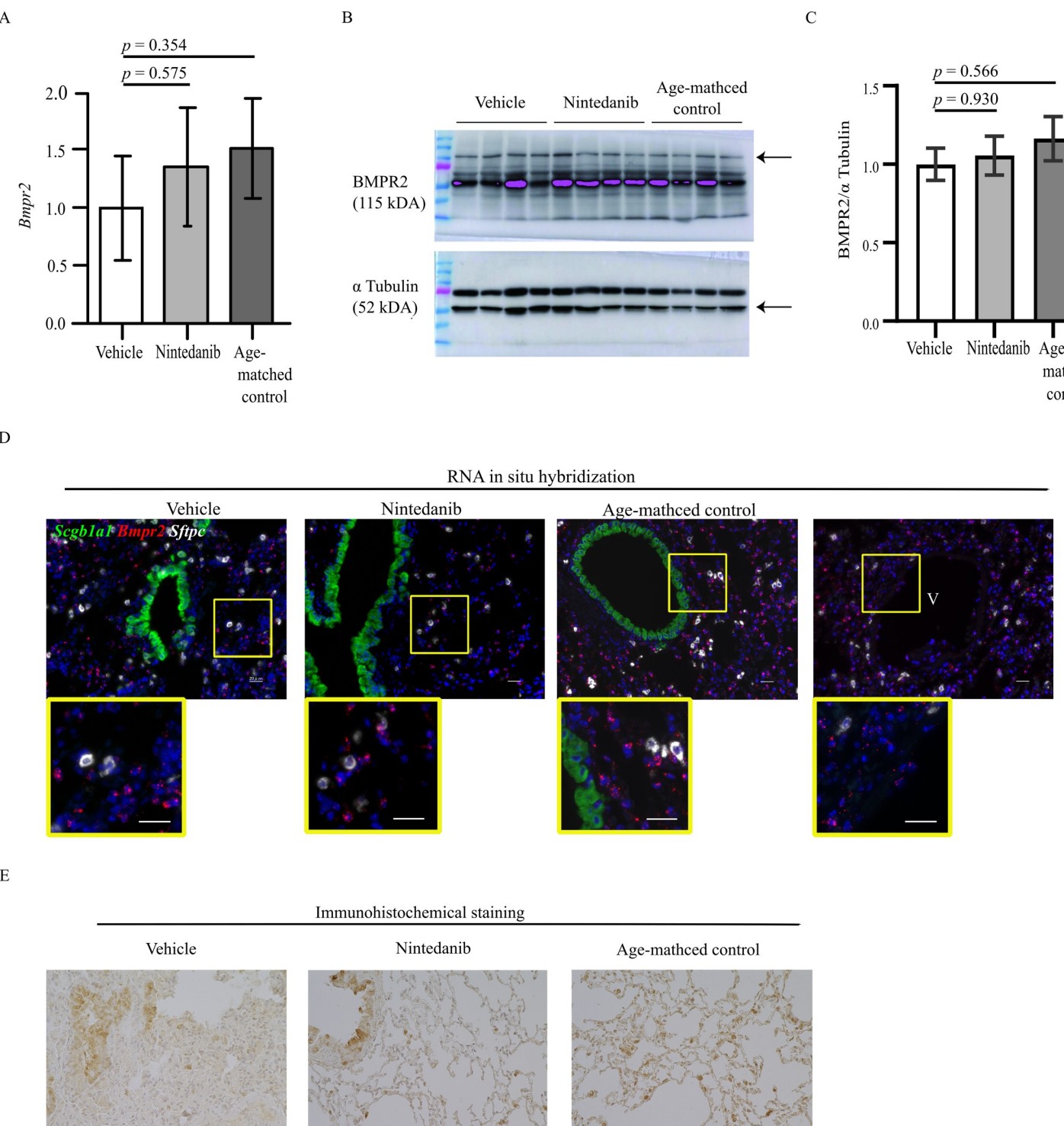

**Fig 5. *Bmpr2* expression in the lung.** (A–C) *Bmpr2* expression in vehicle-treated, nintedanib-treated, and age-matched lung tissue. (A) Quantitative reverse transcription polymerase chain reaction analysis showed that *Bmpr2* expression tended to be downregulated in vehicle-treated mouse lung (n = 3) compared with that in age-matched D1CC×D1BB mouse lung (n = 3) and that this downregulation tended to be attenuated by nintedanib treatment (n = 3). The vehicle- and nintedanib-treated mouse lungs were obtained from the mice used in the analysis shown in Fig 1C. (B, C) Western blot analysis confirmed the findings shown in (A): matched D1CC×D1BB mouse lung, n = 4; vehicle-treated mouse lung, n = 4; nintedanib-treated mouse lung, n = 4 Relative signal intensity was determined using α Tubulin as the loading control. Data are presented as mean ± standard error. (D) RNA in situ hybridization for *Bmpr2* expression in the peripheral lung. *Bmpr2* was expressed mainly in type I alveolar epithelial cells adjacent to type II alveolar epithelial cells with *Sftpc* expression. Scale bars = 20 μm; V, blood vessels. (E) Immunohistochemical staining revealed that BMPR2 localized to some bronchial epithelial cells and alveolar epithelial cells. Scale bars = 20 μm.

the present functional enrichment and pathway analyses may help direct future studies. In addition, nintedanib has recently been reported to have antifibrotic activity in organs other than the lungs, including in mouse models of liver fibrosis and muscular dystrophy [31, 32]; examinations for nintedanib effect in other organs are warranted.

In the present study, fewer upregulated genes than downregulated genes were detected, and among the 3 combinations of nintedanib- and vehicle-treated mice based on the progression of pulmonary fibrosis, only three common upregulated genes were identified (*Rfx2*, *Hspa1a*, and *Hspa1b*; Fig 2A). HSPA1A and HSPA1B are members of the heat shock protein 70 family, and are closely related to one another. Several studies have reported that heat shock proteins play roles in the pathogenesis of interstitial lung disease, including in a bleomycin-induced mouse model of pulmonary fibrosis, in which induction of HSP70 expression inhibited fibrosis in the lung, and administration of anti-HSP70 antibodies reduced lung function and increased mortality [33–35]. In addition, extracellular matrix proteins are reported to be increased in primary lung fibroblasts lacking HSP70 [36]. In the present study, although we examined the changes in the expression levels of *Hspa1a* and *Hspa1b* in nintedanib-treated fibroblasts by means of RT-qPCR, we did not observe any changes in the expression levels of these genes at least under the present conditions.

RFX2 is a DNA-binding protein and a member of the RFX family, which is a protein family that is involved in the cell cycle, immunoregulation, and expression of motile cilia, particularly in spermatogenesis [37–39]. In the airway epithelium of the lung, RFX2 is suggested to coordinate with the transcription factor grainyhead-like 2 to induce differentiation of ciliated cells from progenitor basal cells during epithelial regeneration [40]. Our present immunohistochemical staining results show that RFX2 is expressed mainly in the bronchial epithelium (Fig 5A) and our RNA in situ hybridization results show that *Rfx2* is expressed in cells located adjacent to *Scgb1a1*-positive cells (Fig 5B), indicating that *Rfx2*-positive cells are localized in a basal cell–like position; therefore, nintedanib may affect airway epithelium or secretory cells via *Rfx2* expression in basal cells. In addition, a recent study using single-cell RNA sequencing in pulmonary fibrosis identified pathological *keratin (KRT)5⁻/KRT17⁺* epithelial cells producing extracellular matrix and expressing the canonical basal cell transcriptional factor tumor protein p63 [41]. The *Rfx2* expression–positive cells detected in the present study were located near these cells in areas of basal cells, although *Krt5* showed low mRNA expression before and after nintedanib treatment. *Rfx2* is anticipated to be involved in the repair of pulmonary epithelium during nintedanib treatment; however, it remains unclear how RFX2 functions in the lungs. Although we found no change in the expression of RFX family genes other than *Rfx2*, the regulatory genes *Rfx1–3* have been analyzed by using RNA-seq and ChIP-seq in mouse ependymal cells [42]. In addition, RFX2 is reported to be involved in the regulation of the promoter of *fibroblast growth factor-1B*, a major transcript within the human brain and retina, by forming a complex with RFX3 in human glioblastoma cells, suggesting that RFX2 alone is insufficient for its regulation [43]. Thus, it remains unknown how nintedanib promotes *Rfx2* RNA expression. To answer the question, it will be worthwhile to investigate the relationship between *Rfx2* and non-coding RNAs such as microRNAs. In the present study, the expression of *Mucin5b* (*Muc5b*) was found to be significantly suppressed by nintedanib. A variant promoter of this gene, rs35705950, is reported to be a risk factor for pulmonary fibrosis; furthermore, in pulmonary fibrosis, *Muc5b* has been shown to be expressed not only in the conducting airways but also in epithelial cells lining honeycomb cysts, and its expression level in the mouse bronchoalveolar epithelium is associated with impaired mucociliary clearance and the extent and persistence of bleomycin-induced pulmonary fibrosis [44, 45]. Taken together, the present findings indicate that nintedanib may affect the ciliary phenotypic

expression and differentiation of abnormal cell populations in the peripheral airway, which may underlie its activity to slow the progression of pulmonary fibrosis.

Recently, Calabrese et al. identified 23 upregulated genes in aggregates of fibroblasts lined by epithelial cells, which they called epithelial cell/fibroblastic foci sandwich, in lungs with idiopathic pulmonary fibrosis compared to those in lungs with primary spontaneous pneumothorax by RNA sequencing analysis [46]. Intriguingly, five downregulated genes (*Scgb3a1*, *Bpifb1*, *Pigr*, *Muc5B*, and *Krt5*) by nintedanib treatment in the present study were included in the 14 upregulated genes with log fold-change $\geq 2$ in the 23 upregulated genes. *Scgb3a1*, *Pigr*, *Bpifb1*, and *Muc5B* are related to secretory and mucin proteins, and immune defenses [46]. *Scgb3a1* is expressed in secretory cells in idiopathic pulmonary fibrosis [47]. The BPIFB1 protein was reported to be upregulated in the small airway epithelium in cystic fibrosis compared to that in the control [48]. PIGR mediates immunoglobulin A for immune exclusion of inhaled pathogens in the bronchial mucosa [49]. We previously reported *Krt5* expression in hyperplastic bronchiolar cells in a mouse model of bleomycin-induced interstitial pneumonia [13]. *Muc5B* is reported to be expressed in the small airways and epithelial cells lining honeycomb cysts [44, 45]. These findings imply that these genes could be involved in the development of pulmonary fibrosis and the effect of nintedanib in specific lesions between the peripheral airways and alveoli despite the differences in diseases, species, and sample preparation between the present study and Calabrese's study. Further investigations are needed to clarify the functions of these genes.

BMPR2 is a transmembrane and serine/threonine kinase receptor for bone morphogenetic proteins, which have functions in cell differentiation, proliferation, and bone formation [50]. Bone morphogenetic proteins are thought to affect the vascular endothelium and smooth muscle, interfering with transforming growth factor beta signaling; therefore, mutation in BMPR2 is a cause of pulmonary hypertension [21]. In addition, a previous study reported that BMPR2 expression is reduced in patients with idiopathic pulmonary fibrosis or in those with pulmonary hypertension [51]. Because we hypothesized that this blood vessel–related gene is acted upon by nintedanib via its anti-VEGFR activity, we focused on *Bmpr2* among the genes upregulated by nintedanib treatment. Although *Bmpr2* expression in the epithelium has been reported previously, we confirmed the presence of *Bmpr2* expression in iRA-ILD mouse lung. How nintedanib upregulates *Bmpr2* during the development of lung fibrosis remains to be investigated; however, we speculate that nintedanib may alleviate lung fibrosis through BMPR2 enhancement.

In the present study, *Rfx2* and *Bmpr2* were expressed in peripheral airway epithelium and alveolar cells, as determined by in situ hybridization and immunohistochemistry. Although it is difficult to detect cells with significant gene alteration by using current tissue-based RNA sequencing approaches, we speculate that the responsible cell population may be cells with expression of these genes. Detection of the expression of these genes is difficult in mesenchymal cells such as fibroblasts, especially in nintedanib-treated mice where the development of fibrotic lesions is attenuated by nintedanib treatment. Single-cell sequencing is expected to a useful approach for identifying the cell lineages responsible for the detected alteration of gene expression, and it is expected that it will be specific populations within the small airways or alveolar epithelium that show these changes. Single-cell sequencing has the advantage of being able to be used also for mesenchymal cell analysis, and a previous report has shown that nintedanib treatment in bleomycin-treated mice reduced a bleomycin-induced shift to the extracellular matrix-enriched cluster in fibroblasts [7], although no specific genes of activated cells were detected in pulmonary fibrosis lesions. We used fibroblasts derived from D1CC×D1BC mouse lung to evaluate the effect of nintedanib in fibroblasts; however, fibroblasts cultured after collection from lung tissue may contain several lineages, making it difficult to determine whether the established cells are the actual cells affected by nintedanib.

We acknowledge the following limitations of the present study. First, for the expression analysis, total RNA was extracted from peripheral lung tissue that included not only the alveolar epithelium, stromal tissue, pulmonary blood vessels, and peripheral airways, but also mesenchymal and hematopoietic cells such as macrophages, lymphocytes, and neutrophils. Therefore, we were unable to evaluate the differences in gene expression profiles between discrete pulmonary tissues. We did attempt to determine the specific locations of *Rfx2* and *Bmpr2* expression by using in situ hybridization; however, we could not quantitatively measure the changes in expression because of poor visual discrimination of the signals in specific cell types, structural differences attributed to nintedanib treatment, positioning heterogeneity in various types of hybridized cells. Localized sampling by microdissection or single-cell sequencing may overcome this limitation. In addition, although our methodology allows measurement of gene expression profiles in the periphery of nintedanib-treated lung with pulmonary fibrosis, a consistent means of distinguishing cell types will be needed to determine differential gene expression in the peripheral airway or alveoli. Second, it is difficult to conclude whether the observed up- and downregulation of genes is beneficial or detrimental to pulmonary fibrosis. Also, it remains unknown whether the observed changes in gene expression are directly involved in the antifibrotic activity of nintedanib or whether they reflect changes in response to improved pulmonary fibrosis or are the result of responses related to adverse events associated with nintedanib treatment. Additional studies based on our analysis results may lead to unexpected disadvantageous effects. Because the cellular environment of interstitial lung diseases is controlled by a complex network of interactions among mRNAs, microRNAs, long non-coding RNAs, proteins, and genomic DNA, the iRA-ILD mouse is a valuable mouse model for investigating the expression of genes in the lung environment under antifibrotic treatment.

## Conclusions

Here, we used the iRA-ILD mouse model to examine the effects of nintedanib in the fibrotic lung. The present analyses revealed 157 genes that were up- or downregulated in the lung tissue of iRA-ILD mice by nintedanib treatment. Subsequent functional enrichment analysis revealed that the identified genes were associated with extracellular components, including the myocardial architecture. Of the upregulated genes, we focused on *Rfx2* and *Bmpr2*. *Rfx2* was highly expressed in cells adjacent to *Scgb1a1*-positive cells localized in a basal cell–like position, indicating that nintedanib treatment has some effect on the peripheral airway epithelium. *Bmpr2* may be involved in the development of pulmonary fibrosis in the alveolar region. As it is difficult to collect human lung tissue for comprehensive gene expression analysis in clinical practice, this mouse model will be valuable for tracking fibrosis and treatment responsiveness over a relatively long period of medication. Our findings are expected to contribute to the development of improved strategies for the use of nintedanib for the treatment of devastating interstitial lung diseases.

## Supporting information

**S1 Raw images.**
(TIF)

**S2 Raw images.**
(TIF)

## Acknowledgments

We thank Dr. M. Murata for his cooperation in measuring serum SP-D levels. We also thank Boehringer Ingelheim Pharma GmbH and Co. KG for providing nintedanib used in the study.

## Author Contributions

**Conceptualization:** Shintaro Mikami, Yoko Miura, Shinji Kondo, Hideki Noguchi, Satoshi Kanazawa, Kazutsugu Uematsu.

**Data curation:** Shintaro Mikami, Yoko Miura, Shinji Kondo, Hideki Noguchi, Satoshi Kanazawa.

**Formal analysis:** Shintaro Mikami, Yoko Miura, Shinji Kondo, Hideki Noguchi, Satoshi Kanazawa.

**Funding acquisition:** Satoshi Kanazawa, Kazutsugu Uematsu.

**Investigation:** Shintaro Mikami, Yoko Miura, Kosuke Sakai, Hiroaki Nishimura, Hiroyuki Kyoyama, Gaku Moriyama, Nobuyuki Koyama, Hirotsugu Ohkubo, Satoshi Kanazawa.

**Methodology:** Shintaro Mikami, Yoko Miura, Shinji Kondo, Hideki Noguchi, Satoshi Kanazawa, Kazutsugu Uematsu.

**Project administration:** Kazutsugu Uematsu.

**Resources:** Shintaro Mikami, Yoko Miura, Kosuke Sakai, Hiroaki Nishimura, Hiroyuki Kyoyama, Gaku Moriyama, Nobuyuki Koyama, Hirotsugu Ohkubo, Satoshi Kanazawa.

**Supervision:** Satoshi Kanazawa, Kazutsugu Uematsu.

**Visualization:** Shintaro Mikami, Yoko Miura, Shinji Kondo, Satoshi Kanazawa.

**Writing – original draft:** Shintaro Mikami.

**Writing – review & editing:** Yoko Miura, Shinji Kondo, Hideki Noguchi, Satoshi Kanazawa, Kazutsugu Uematsu.

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
