## [Decision Letter · Decision Letter 0]

9 Feb 2022

PONE-D-22-01668Nintedanib induces gene expression changes in the lung of induced-rheumatoid arthritis–associated interstitial lung disease micePLOS ONE

Dear Dr. Uematsu,

Thank you for submitting your manuscript to PLOS ONE. After careful consideration, we feel that it has merit but does not fully meet PLOS ONE’s publication criteria as it currently stands. Therefore, we invite you to submit a revised version of the manuscript that addresses the points raised during the review process. Our reviewers found some interests in this manuscript, but also pointed out a number of criticisms that require improvement and amendment. I ask the authors to fully respond to all comments made by reviewers in the revised manuscript. This includes additional experiments. 

We look forward to receiving your revised manuscript.

Kind regards,

Masataka Kuwana, MD, PhD

Academic Editor

PLOS ONE

Journal Requirements:

2. As part of your revision, please complete and submit a copy of the Full ARRIVE 2.0 Guidelines checklist, a document that aims to improve experimental reporting and reproducibility of animal studies for purposes of post-publication data analysis and reproducibility: https://arriveguidelines.org/sites/arrive/files/Author%20Checklist%20-%20Full.pdf (PDF). Please include your completed checklist as a Supporting Information file. Note that if your paper is accepted for publication, this checklist will be published as part of your article.

3. Thank you for stating the following in the Competing Interest section: 

"KU, SK, and HO received research support from Boehringer Ingelheim Pharma GmbH & Co. KG. All other authors have nothing to disclose."

We note that you received funding from a commercial source: Boehringer Ingelheim Pharma GmbH & Co. KG. Please provide an amended Competing Interests Statement that explicitly states this commercial funder, along with any other relevant declarations relating to employment, consultancy, patents, products in development, marketed products, etc. 

Reviewers' comments:

Reviewer's Responses to Questions

**Comments to the Author**

1. Is the manuscript technically sound, and do the data support the conclusions?

Reviewer #1: No

Reviewer #2: Yes

Reviewer #3: Partly

2. Has the statistical analysis been performed appropriately and rigorously? 

Reviewer #1: Yes

Reviewer #2: Yes

Reviewer #3: Yes

3. Have the authors made all data underlying the findings in their manuscript fully available?

Reviewer #1: Yes

Reviewer #2: Yes

Reviewer #3: Yes

4. Is the manuscript presented in an intelligible fashion and written in standard English?

Reviewer #1: Yes

Reviewer #2: Yes

Reviewer #3: Yes

5. Review Comments to the Author

Reviewer #1: The authors examined the alteration of gene expression in murine lung fibrosis by the treatment with nintedanib. A number of criticisms are raised for this study.

1) The authors need to show histological images of the lung, H&E and Masson trichrome stains. Also, they need to show if the treatment with nintedanib affected the interstitial lung disease.

2) In figure 1, they selected 2 vehicle-treated samples from total 3, and 3 nintedanib-treated samples from total 5. The authors should describe if it is rational. Why the other samples showed different gene expression even with nintedanib treatment? Lung histology was also different?

3) In figure 5, Rfx2 and Bmpr2 are expressed in type II alveolar cells or endothelial cells. However, in figure 4 they analyzed the expression of fibroblast in vitro. They need to show if the genes express in fibroblast in the lung tissue. Also, they need to show if the gene and protein expressions was increased in lung tissue by the treatment with nintedanib.

Reviewer #2: In this manuscript by Mikami et al. investigated the expression of genes in the lungs of nintedanib-treated mouse model of RA-ILD by next-generation RNA sequencing. The authors identified upregulated and downregulated genes, and the functions and pathways enriched for the differentially expressed genes in nintedanib-treated lungs. Although the statistical analysis of the data is of overall good quality, there are several concerns regarding this study that the authors need to clarify.

The use of various control is lacking in the current study. How is the expression of Rfx2 and Bmr2 in the lungs of control D1CC×D1BC mice compared to iRA-ILD mice? In addition, the inclusion of control (normal lung of D1CC×D1BC mice) would be helpful for the experiment in Fig. 5.

The main conclusions are based on about 2-fold changes in mRNA expression with a small number of samples. Figure 4A is lacking control and any quantification as to whether the expression of RFX2 and BMPR2 changes with nintedanib. Also, the protein levels of RFX2 and BMPR2 in fibroblast should be quantified by Western blot in Fig. 4B.

All the images should be more carefully assessed to demonstrate some degree of quantitative localization to RFX2 and BMPR2. In Fig. 5, the reviewer cannot see any difference in expression and distribution of the genes between vehicle and nintedanib group. In addition, the immunohistochemical staining for Bmpr2 is lacking in Fig. 5.

Reviewer #3: Comments to the authors:

In this study, the authors evaluated the effects of nintedanib on gene expression in the lung of iRA-ILD model mice and revealed Rfx2, Bmpr2 were upregulated by nintedanib. The authors mentioned that the purpose of this study was to identify predictors of nintedanib treatment response or adverse events associated with nintedanib, but current data only showed the changes of the genes by nintedanib without underlying mechanisms. Specific comments are listed below:

Major comments

First of all, all the figures including this article are unclear, and I cannot evaluate the data correctly. The authors should reupload more clear images that meet submission guidelines.

In this article, the authors performed whole lung RNA-seq, but there are many cell types in the entire lung, and it is quite difficult to point out the cell specificity of the identified genes. Therefore, to reveal predictors of nintedanib treatment response, to determine the cells affected by nintedanib are essential, and cell-type-specific gene expression analysis or single-cell RNA sequence is necessary to evaluate the mechanism of nintedanib

Since one scRNA-sequencing study using a bleomycin-induced lung fibrosis model has already been published in 2019 (Peyser et al., Am J Respir Cell Mol Biol. 2019), the authors should mention this in the discussion.

The authors found increased Rfx2, Bmpr2 expressions in the lung of iRA-ILD model mice treated with nintedanib compared with vehicle controls but did not show any functional significance of this finding. More detailed in vitro experiments will be needed to figure out the mechanisms.

Minor points

1) In isolation and culture of lung fibroblast method, lung fibroblasts were cultured with DMEM(1% FBS) for 24h as control, DMEM (1% FBS) with PDGF-BB 10ng/ml for 24h as PDGF-BB condition, and DMEM (1% FBS) containing PDGF-BB 10ng/ml for 24h and add 1 uM nintedanib for 24h as PDGF→nintedanib condition. In 3 conditions, the total culture hours seem different, which might influence their findings due to cell viability.

2) The authors described the Rfx2 and Bmpr2 genes with significant differences in RNA-sequencing data, but they didn’t show the RT-qPCR data using the same samples or same model samples to confirm the RNA-sequencing data. These secured data or additional experiments using the same mice model to verify the Rfx2, Bmpr2 gene expression should be added.

3)The authors used fibroblasts derived from non-fibrotic mice lungs with PDGF-BB stimulation to reveal Rfx2, Bmpr2 gene expression, even if their enrichment analysis and pathway analysis didn’t tell “signal pathway” or “focal adhesion” associated PDGF. The authors should explain the particular reason for using PDGF-BB.

In addition, both Rfx2, Bmpr2 gene expressions didn’t alter by PDGF-BB stimulation, suggesting these two gene expressions may not be regulated by the PDGF signaling pathway.

4) In figure 4A, the authors should show isotype control data in addition to 3 different conditions, control, PDGF-BB, and PDGF-BB→nintedanib, similar to in figure 4B, to see the changes by nintedanib treatment.

5) In figure 5C, the authors revealed Bmpr2 was expressed mainly in type I alveolar epithelial cells, but some expression was also observed in type II alveolar epithelial and vascular endothelial cells by in situ hybridization. They also showed Bmpr2 expression in lung fibroblast in figure 4A, indicating nintedanib may affect multiple types of cells. Detailed functional analyses are necessary to r

6. PLOS authors have the option to publish the peer review history of their article (what does this mean?). If published, this will include your full peer review and any attached files.

Reviewer #1: No

Reviewer #2: No

Reviewer #3: No

---

## [Author Response · Author response to Decision Letter 0]

26 Mar 2022

Responses to Reviewer 1

The authors examined the alteration of gene expression in murine lung fibrosis by the treatment with nintedanib. A number of criticisms are raised for this study.

1) The authors need to show histological images of the lung, H&E and Masson trichrome stains. Also, they need to show if the treatment with nintedanib affected the interstitial lung disease.

(Response) Our coauthors Y. Miura, H. Ohkubo, and S. Kanazawa have previously reported the effects of nintedanib on pulmonary fibrosis in the same mouse model as that used in the present study (ref. 12 in the revised manuscript). We have also conducted an additional histological examination of vehicle-treated and nintedanib-treated lungs using hematoxylin and eosin and Masson’s trichrome staining (new Fig 1A, B) and found less disruption of the pulmonary architecture and less Masson staining in the nintedanib-treated lung than in the vehicle-treated lung. The methods and data from this examination have been added to the revised manuscript as follows:

• Materials and methods, page 5, lines 107–109

We have added “Lungs were fixed in 4% paraformaldehyde, and 2-μm-thick paraffin sections were stained with hematoxylin and eosin and Masson’s trichrome.”

• Results, page 9, lines 230–232

We have added “Histological examination revealed that nintedanib treatment alleviated destruction of the alveolar structure (Fig 1A: hematoxylin and eosin staining) and reduced the density of fibrotic regions (Fig 1B: Masson’s trichrome staining), which is consistent with a previous report [12].”

• Figure 1 legend

We have added “(A, B) Histology of vehicle- and nintedanib-treated lung.”

2) In figure 1, they selected 2 vehicle-treated samples from total 3, and 3 nintedanib-treated samples from total 5. The authors should describe if it is rational. Why the other samples showed different gene expression even with nintedanib treatment? Lung histology was also different?

(Response) We examined pulmonary fibrosis progression by using a plot of Masson staining blue ratio vs. serum SP-D (new Fig 1C) and found clear variation within the two groups of vehicle- and nintedanib-treated mice. Among the three vehicle-treated lungs, one (VC) showed different pulmonary fibrosis progression compared with the other two (VA and VB). Similarly, among the five nintedanib-treated lungs, three (NA, NB, and NC) showed different pulmonary fibrosis progression compared with the other two (NE and ND). The implications here are that: VA and VB had more progressive fibrosis in the lung compared with VC; NA, NB, and NC had more alleviation in interstitial lung disease by nintedanib treatment compared with ND and NE, and by extension that VC was more resistant to the development of severe interstitial lung disease than VA and VB; and ND and NE were less sensitive to nintedanib treatment than were NA, NB, and NC. Therefore, we excluded VC, ND, and NE from the later analyses.

Subsequent gene expression analysis by RNA sequencing using mRNA derived from the three vehicle-treated lungs and five nintedanib-treated lungs also revealed the presence of two major clusters: one comprising the vehicle-treated lungs VA and VB, and one comprising the nintedanib-treated lungs NA, NB, and NC, which supports the findings described above.

We have made the following changes in the revised manuscript:

• Results, pages 9–10, lines 220–225

“Masson staining blue ratio was used an index of the level of fibrosis. Plotting the relationship between Masson stain blue ratio and serum SP-D level revealed a cluster of three of the nintedanib-treated mice (NA, NB, and NC), which was located far from two of the vehicle-treated mice (VA and VB) (Fig 1A).” has been changed to “Next, using Masson staining blue ratio as an index of the level of fibrosis, we plotted the relationship between Masson stain blue ratio and serum SP-D level for 3 vehicle-treated (VA–VC) and 5 nintedanib-treated (NA–NE) iRA-ILD mice. This plot revealed a cluster of three of the nintedanib-treated mice (NA, NB, and NC) that was located far from the two vehicle-treated mice (VA and VB) (Fig 1C).”

3) In figure 5, Rfx2 and Bmpr2 are expressed in type II alveolar cells or endothelial cells. However, in figure 4 they analyzed the expression of fibroblast in vitro. They need to show if the genes express in fibroblast in the lung tissue. Also, they need to show if the gene and protein expressions was increased in lung tissue by the treatment with nintedanib.

(Response) The expressions of Rfx2 and Bmpr2 were detected in peripheral bronchial epithelium and alveolar cells by means of immunohistochemical staining and RNA in situ hybridization. However, it was difficult to evaluate these expressions in lung tissue fibroblasts, partly because of attenuation of fibrosis in nintedanib-treated lung tissue with enhanced expression of Rfx2 and Bmpr2 in the tissue-based RNA sequencing. Instead, RT-qPCR and western blotting analyses revealed that the expression of both genes tended to be downregulated in iRA-ILD mouse lung compared with that in age-matched D1CC×D1BB mouse lung and that this downregulation tended to be ameliorated by nintedanib treatment. We believe that these data provide further evidence that nintedanib treatment alters Rfx2 and Bmpr2 gene expression. We currently attribute the differential expressions of the two genes to the alteration of gene expression in peripheral bronchial epithelium or alveolar cells rather than in fibroblasts. It is difficult to evaluate involvement of fibroblasts in expression changes of both genes. We think that fibroblasts isolated from lung tissue contain various lineages; therefore, sorting by receptors such as PDGFR may be needed to examine this issue further.

We have made the following changes in the revised manuscript:

• Figures 4 and 5

We have added Rfx2 and Bmpr2 expression data for vehicle-treated, nintedanib-treated, and age-matched mouse lung, as determined by means of RT-qPCR and western blot analysis.

• Materials and methods, pages 7–8, lines 170–180

We have added our western blot procedures.

• Results, page 21, lines 304–307

We have added the text “RT-qPCR and western blotting analyses showed that the expression of Rfx2 was significantly downregulated in the iRA-ILD lung compared with that in the age-matched D1CC×D1BB lung, and that this downregulation tended to be ameliorated by nintedanib treatment (Fig 4A–C).”

• Results, pages 21–22, lines 316–319

We have added the text “RT-qPCR and western blotting analyses also showed that Bmpr2 expression tended to be downregulated in the vehicle-treated lung compared with that in the age-matched D1CC×D1BB lung, and that this downregulation also tended to be ameliorated by nintedanib treatment (Fig 5A–C).”

Responses to Reviewer 2

In this manuscript by Mikami et al. investigated the expression of genes in the lungs of nintedanib-treated mouse model of RA-ILD by next-generation RNA sequencing. The authors identified upregulated and downregulated genes, and the functions and pathways enriched for the differentially expressed genes in nintedanib-treated lungs. Although the statistical analysis of the data is of overall good quality, there are several concerns regarding this study that the authors need to clarify.

The use of various control is lacking in the current study. How is the expression of Rfx2 and Bmr2 in the lungs of control D1CC×D1BC mice compared to iRA-ILD mice? In addition, the inclusion of control (normal lung of D1CC×D1BC mice) would be helpful for the experiment in Fig. 5.

(Response) As requested, we have repeated the experiments in D1CC×D1BC mice. RT-qPCR and western blot analysis showed higher expressions of Rfx2 and Bmpr2 in D1CC×D1BC lung compared with those in iRA-ILD lung.

We have made the following changes in the revised manuscript:

• Figures 4 and 5

We have added Rfx2 and Bmpr2 expression data for vehicle-treated, nintedanib-treated, and age-matched mouse lung, as determined by means of RT-qPCR and western blot analysis. Immunohistochemical staining and in situ hybridization data have also been added.

• Materials and methods, pages 6, lines 130–131

We have added the phrase “and 4 age-matched D1CC×D1BC mice,”.

• Materials and methods, pages 7, lines 165–166

We have added the words “the age-matched”.

• Materials and methods, pages 7–8, lines 170–180

We have added our western blot procedures.

The main conclusions are based on about 2-fold changes in mRNA expression with a small number of samples. Figure 4A is lacking control and any quantification as to whether the expression of RFX2 and BMPR2 changes with nintedanib. Also, the protein levels of RFX2 and BMPR2 in fibroblast should be quantified by Western blot in Fig. 4B.

(Response) As requested, we evaluated Rfx2 and Bmpr2 expressions in age-matched D1CC×D1BC mouse lung by RT-qPCR and western blot analysis and compared the data with those obtained from iRA-ILD mouse lung. Rfx2 and Bmpr2 expression were both downregulated in iRA-ILD lung compared to that in age-matched D1CC×D1BC lung, and nintedanib treatment tended to ameliorate these downregulations. Immunohistochemical staining and RNA in situ hybridization analyses revealed that Rfx2 and Bmpr2 were expressed in peripheral bronchial epithelium and alveolar cells. However, it was difficult to examine the expression of Rfx2 and Bmpr2 in lung tissue fibroblasts partly due to the attenuation of fibrosis in nintedanib-treated lung with enhanced expression of Rfx2 and Bmpr2 in the tissue-based sequencing. We currently attribute the differential gene expression to the alteration of gene expression in peripheral bronchial epithelium or alveolar cells rather than in fibroblasts. We think that because fibroblasts isolated from lung tissue contain various lineages, sorting by receptors such as PDGFR will be needed to examine this issue further. We are currently conducting a comprehensive analysis of gene expression changes induced by nintedanib treatment, and we have identified several additional genes of interest other than Rfx2 and Bmpr2. Therefore, because the present examination of fibroblasts needs further quantification, we have removed those experiments from the revised manuscript, and we intend to report our findings for our other genes of interest together with the changes of Rfx2 and Bmpr2 expression in fibroblasts in a future report.

We have made the following changes in the revised manuscript:

• Figures 4 and 5

We have added Rfx2 and Bmpr2 expression data for vehicle-treated, nintedanib-treated, and age-matched mouse lung, as determined by means of RT-qPCR and western blot analysis.

• Abstract, page 2, lines 34–36; Materials and methods, page 5, lines 115–126 and page 8, 183–193; Results, page 21, lines 307–309 and page 26, lines, 440–441

We have deleted text discussing fibroblasts.

• Discussion, line 469

The phrase “the upregulation of” has been deleted.

• Discussion, line 470

The phrase “in fibroblasts following nintedanib treatment” has been deleted.

• Conclusion, lines 518–519

The phrase “which were found to be expressed in lung fibroblasts” has been deleted.

All the images should be more carefully assessed to demonstrate some degree of quantitative localization to RFX2 and BMPR2. In Fig. 5, the reviewer cannot see any difference in expression and distribution of the genes between vehicle and nintedanib group. In addition, the immunohistochemical staining for Bmpr2 is lacking in Fig. 5.

(Response) As requested, we have added images of immunohistochemical staining of BMPR2 expression as new Figure 5(E). We also evaluated Rfx2 and Bmpr2 expression in age-matched D1CC×D1BC mouse lung by means of immunohistochemistry and in situ hybridization and found no differences in distribution among vehicle-, nintedanib-treated, and age-matched D1CC×D1BC mouse lung. 

Responses to Reviewer 3

In this study, the authors evaluated the effects of nintedanib on gene expression in the lung of iRA-ILD model mice and revealed Rfx2, Bmpr2 were upregulated by nintedanib. The authors mentioned that the purpose of this study was to identify predictors of nintedanib treatment response or adverse events associated with nintedanib, but current data only showed the changes of the genes by nintedanib without underlying mechanisms. Specific comments are listed below:

Major comments

First of all, all the figures including this article are unclear, and I cannot evaluate the data correctly. The authors should reupload more clear images that meet submission guidelines.

(Response) We have prepared the figures in line with the PLOS ONE submission guidelines and confirmed the quality of the figures through PACE, which PLOS ONE recommends for assessment of figure quality. We will consult the editorial office if the issue persists.

In this article, the authors performed whole lung RNA-seq, but there are many cell types in the entire lung, and it is quite difficult to point out the cell specificity of the identified genes. Therefore, to reveal predictors of nintedanib treatment response, to determine the cells affected by nintedanib are essential, and cell-type-specific gene expression analysis or single-cell RNA sequence is necessary to evaluate the mechanism of nintedanib

Since one scRNA-sequencing study using a bleomycin-induced lung fibrosis model has already been published in 2019 (Peyser et al., Am J Respir Cell Mol Biol. 2019), the authors should mention this in the discussion.

(Response) As the reviewer notes, it is difficult to determine the cell specificity of the identified genes. However, the aim of the present study was to obtain information about a potential predictor of nintedanib response by using lung tissue from a model mouse. We intend to continue this line of research by examining cell specificity after identifying which genes expressions are altered.

At present, we have evaluated Rfx2 and Bmpr2 expression by means of RNA in situ hybridization and immunohistochemistry and these examinations showed that these genes are expressed in peripheral airway epithelium and alveolar cells. Looking forward, single-cell sequencing may be a useful method for identifying alterations of gene expression in specific cells in lung lesions. As noted by the reviewer, Peyser et al. have utilized this approach to elucidate the function of fibroblasts in a bleomycin-induced pulmonary fibrosis model. In their study, treatment with nintedanib for 11 days in bleomycin-treated mice reduced bleomycin-induced shift to the ECM-enriched cluster in fibroblasts, and these nintedanib-treated fibroblasts tended to have reduced expressions of Col1a1, Col3a1, and Fn1. They also showed that no specific genes were activated in the cells in pulmonary fibrosis lesions.

In our study, we used iRA-ILD model mice and we adopted a longer period of nintedanib treatment (8 weeks) than that used by Peyser et al. After we identified two genes of interest, Rfx2 and Bmpr2, we confirmed downregulation of both gene expressions in iRA-ILD mouse lung compared with that in D1CC×D1BC mouse lung without ILD induction, and the tendency of these downregulations to be ameliorated by nintedanib treatment, by means of RT-qPCR and western blot analysis; however, at present, we cannot determine which specific cells contributed to this differential gene expression. Although tissue-based RNA sequencing, as used in the present study, may mask differences of gene expression in specific or small populations of cells, we currently attribute the differential expressions of Rfx2 and Bmpr2 to the alteration of gene expression in hybridized cell populations by immunohistochemistry or RNA in situ hybridization of Rfx2 and Bmpr2. However, the immunohistochemical staining and RNA in situ hybridization analyses did not clearly show Rfx2 or Bmpr2 expression in fibroblasts in the lung tissues, even though immunostaining and RT-qPCR analyses showed Rfx2 or Bmpr2 expression in cultured lung fibroblasts. Fibroblasts contain several lineages; therefore, our fibroblast data needs to be confirmed under various conditions.

We have made the following changes in the revised manuscript:

• Introduction, pages 3–4, lines 70–74

“Peyser et al. used single-cell sequencing and a bleomycin-induced pulmonary fibrosis model to characterize molecular response to fibrotic injury [7]. In that study, treatment of nintedanib reduced the bleomycin-induced cluster shift of fibroblasts to the extracellular matrix-enriched cluster and no specific upregulation of gene expression was observed in pulmonary fibrosis lesion cells.”

• Introduction, page 4, lines 74–75

“However, this approach is yet to be used to examine the gene expression alterations in nintedanib-treated lung.” has been changed to “A tissue-based sequencing approach has yet to be used to examine the gene expression alterations in nintedanib-treated lung.”

• Discussion, pages27–28, lines 473–489

“In the present study, Rfx2 and Bmpr2 were expressed in peripheral airway epithelium and alveolar cells, as determined by in situ hybridization and immunohistochemistry. Although it is difficult to detect cells with significant gene alteration by using current tissue-based RNA sequencing approaches, we speculate that the responsible cell population may be cells with expression of these genes. Detection of the expression of these genes is difficult in mesenchymal cells such as fibroblasts, especially in nintedanib-treated mice where the development of fibrotic lesions is attenuated by nintedanib treatment. Single-cell sequencing is expected to a useful approach for identifying the cell lineages responsible for the detected alteration of gene expression, and it is expected that it will be specific populations within the small airways or alveolar epithelium that show these changes. Single-cell sequencing has the advantage of being able to be used also for mesenchymal cell analysis, and a previous report has shown that nintedanib treatment in bleomycin-treated mice reduced a bleomycin-induced shift to the extracellular matrix-enriched cluster in fibroblasts [7], although no specific genes of activated cells were detected in pulmonary fibrosis lesions. We used fibroblasts derived from D1CC×D1BC mouse lung to evaluate the effect of nintedanib in fibroblasts; however, fibroblasts cultured after collection from lung tissue may contain several lineages, making it difficult to determine whether the established cells are the actual cells affected by nintedanib.”

The authors found increased Rfx2, Bmpr2 expressions in the lung of iRA-ILD model mice treated with nintedanib compared with vehicle controls but did not show any functional significance of this finding. More detailed in vitro experiments will be needed to figure out the mechanisms.

(Response) Our iRA-ILD mouse model develops ILD slowly which is less acute compared to bleomycin-induced ILD. In our model, we found altered expression of Rfx2 and Bmpr2 during a comparatively longer treatment of nintedanib. In the airway epithelium, RFX2 coordinates with the transcription factor grainhead-like 2 to induce differentiation of ciliate cells from progenitor basal cells during epithelial regeneration (Gao X, et al. J Cell Biol, 2015;211:669-682). Immunohistochemical staining showed that RFX2 was expressed mainly in bronchial epithelium, and RNA in situ hybridization showed that Rfx2 is expressed in cells located adjacent to Scgb1a1-positive cells, which are proposed to be club cells. From these findings, we speculate that Rfx2 is related to repair of the small airways and lung epithelium in response to disruption of the lung architecture; however, more data on RFX2 expression data under various pathological conditions are needed to confirm this role.

Bmpr2 mutation is reported to be associated with pulmonary hypertension, and it has been reported that BMPR2 expression is reduced in patients with idiopathic pulmonary fibrosis or pulmonary hypertension (Chen NY, et al. Am J Physiol Lung Cell Mol Physiol. 2016;311:L238-254). How BMPR2 is related to alleviation of pulmonary fibrosis remains to be fully investigated. However, we confirmed BMPR2 expression in fibroblasts and lung epithelium and we expect BMPR2 to exert its function in these cells and tissues.

Minor points

1) In isolation and culture of lung fibroblast method, lung fibroblasts were cultured with DMEM(1% FBS) for 24h as control, DMEM (1% FBS) with PDGF-BB 10ng/ml for 24h as PDGF-BB condition, and DMEM (1% FBS) containing PDGF-BB 10ng/ml for 24h and add 1 uM nintedanib for 24h as PDGF→nintedanib condition. In 3 conditions, the total culture hours seem different, which might influence their findings due to cell viability.

(Response) Originally, we used four culture conditions and evaluated Rfx2 and Bmpr2 expressions in fibroblasts by RT-qPCR. The four conditions were as follows:

1) culture in DMEM (0.1% FBS) for 24 h,

2) culture in DMEM (0.1% FBS) containing PDGF-BB for 24 h,

3) culture in DMEM (0.1% FBS) containing PDGF-BB for 24 h followed by culture in DMEM (0.1% FBS) with 1 μM nintedanib for 24 h (i.e., PDGF-BB → with nintedanib)

4) culture in DMEM (0.1% FBS) containing PDGF-BB for 24 h followed by culture in DMEM (0.1% FBS) without 1 μM nintedanib which was dissolved in tepid water, for 24 h (i.e., PDGF-BB → without nintedanib)

Cell viability was not affected by the addition of 1 μM of nintedanib. Fibroblasts cultured under the (PDGF-BB → with nintedanib) condition showed significantly higher expressions of Rfx2 and Bmpr2 compared with those in fibroblasts cultured under the (PDGF-BB → without nintedanib) condition. Fibroblasts cultured under the (PDGF-BB → without nintedanib) condition showed the same expressions of Rfx2 and Bmpr2 as fibroblasts cultured in DMEM (0.1% FBS) containing PDGF-BB for 24 h. These data are presented in the figure below.

2) The authors described the Rfx2 and Bmpr2 genes with significant differences in RNA-sequencing data, but they didn’t show the RT-qPCR data using the same samples or same model samples to confirm the RNA-sequencing data. These secured data or additional experiments using the same mice model to verify the Rfx2, Bmpr2 gene expression should be added.

(Response) Additional RT-qPCR and western blot analyses showed that expression of both genes tended to be downregulated in iRA-ILD lung compared with that in age-matched D1CC×D1BB lung, and that this downregulation tended to be ameliorated by nintedanib treatment.

We have made the following changes in the revised manuscript:

• Figures 4 and 5

We have added Rfx2 and Bmpr2 expression data for vehicle-treated, nintedanib-treated, and age-matched mouse lung, as determined by means of RT-qPCR and western blot analysis.

• Materials and methods, pages 7–8, lines 170–180

We have added our western blotting procedures.

• Results, page 21, lines 304–307

We have added the text “RT-qPCR and western blotting analyses showed that the expression of Rfx2 was significantly downregulated in the vehicle-treated lung compared with that in the age-matched D1CC×D1BB lung and that this downregulation tended to be ameliorated by nintedanib treatment (Fig 4A–C).”

3)The authors used fibroblasts derived from non-fibrotic mice lungs with PDGF-BB stimulation to reveal Rfx2, Bmpr2 gene expression, even if their enrichment analysis and pathway analysis didn’t tell “signal pathway” or “focal adhesion” associated PDGF. The authors should explain the particular reason for using PDGF-BB.

(Response) We used PDGF-BB because it binds to PDGF receptor ��, ��, and �� on mesenchymal cells, activates fibroblasts, and is reported to be involved in pathogenesis of fibrotic diseases (Bonner JC. Regulation of PDGF and its receptors in fibrotic diseases. Cytokine Growth Factor Rev. 2004;15:255-273; Hoyle GW, et al. Emphysematous lesions, inflammation, and fibrosis in the lungs of transgenic mice overexpressing platelet-derived growth factor. Am J Pathol. 1999;154:1763-1775; Karakiulakis G, et al. Cell type-specific effect of hypoxia and platelet-derived growth factor-BB on extracellular matrix turnover and its consequences for lung remodeling. J Biol Chem. 2007;282:908-915).

In addition, both Rfx2, Bmpr2 gene expressions didn’t alter by PDGF-BB stimulation, suggesting these two gene expressions may not be regulated by the PDGF signaling pathway.

(Response) RT-qPCR and western blot analyses showed that Rfx2 and Bmpr2 expressions were reduced in iRA-ILD lung compared with those in D1CC×D1BC lung. ILD develops slowly in iRA-ILD mice, and PDGF-BB is suspected to function mildly compared with that in acutely progressive ILD. Rfx2 and Bmpr2 expressions were downregulated during slow ILD development in our model mice; therefore, we think that neither gene may be regulated by acute stimulation of PDGF-BB. PDGF-BB activates various genes via PDGF–PDGFR signaling, and nintedanib, an inhibitor of PDGFR, FGFR, and VEGFR, is expected to inhibit fibroblast activation through several signaling pathways. Therefore, Rfx2 and Bmpr2 expression in fibroblasts may be affected indirectly through one or more of these pathways.

4) In figure 4A, the authors should show isotype control data in addition to 3 different conditions, control, PDGF-BB, and PDGF-BB→nintedanib, similar to in figure 4B, to see the changes by nintedanib treatment.

(Response) Nintedanib is a small-molecule tyrosine kinase inhibitor that dissolves in tepid water; therefore, we used water as the control.

5) In figure 5C, the authors revealed Bmpr2 was expressed mainly in type I alveolar epithelial cells, but some expression was also observed in type II alveolar epithelial and vascular endothelial cells by in situ hybridization. They also showed Bmpr2 expression in lung fibroblast in figure 4A, indicating nintedanib may affect multiple types of cells. Detailed functional analyses are necessary to r

(Response) In our study, we found that Bmpr2 is expressed in type I and II alveolar epithelial cells, vascular endothelial cells, and fibroblasts. The IPF Cell Atlas (http://www.ipfcellatlas.com/), a collection of several single-cell RNA sequencing datasets related to IPF in multi-institutional collaboration published by the Kaminski Lab, also shows BMPR2 expression in these cells, as well as some expression in ciliated, club, and basal cells. BMPR2 mutation is reported to be associated with pulmonary hypertension, and it is reported that BMPR2 expression is reduced in patients with idiopathic pulmonary fibrosis or pulmonary hypertension (Chen NY, et al. Am J Physiol Lung Cell Mol Physiol. 2016;311:L238-254). We think that BMPR2 may function in alveolar epithelium and other vessels for the repair of the pulmonary architecture during ILD progression; however, we agree with the reviewer that further investigations of the function of BMPR2 are needed.

Error correction:

• Figure 2 B

“NC vs, VA” has been changed to “NC vs. VA”.

---

## [Decision Letter · Decision Letter 1]

18 Apr 2022

PONE-D-22-01668R1Nintedanib induces gene expression changes in the lung of induced-rheumatoid arthritis–associated interstitial lung disease micePLOS ONE

Dear Dr. Uematsu,

Thank you for submitting your manuscript to PLOS ONE. After careful consideration, we feel that it has merit but does not fully meet PLOS ONE’s publication criteria as it currently stands. Therefore, we invite you to submit a revised version of the manuscript that addresses the points raised during the review process.

The manuscript has been improved by revisions, but one of the reviewers pointed out some major concerns that have not been adequately answered. These points are important and require further improvement. 

We look forward to receiving your revised manuscript.

Kind regards,

Masataka Kuwana, MD, PhD

Academic Editor

PLOS ONE

Journal Requirements:

Additional Editor Comments (if provided):

Reviewers' comments:

Reviewer's Responses to Questions

**Comments to the Author**

1. If the authors have adequately addressed your comments raised in a previous round of review and you feel that this manuscript is now acceptable for publication, you may indicate that here to bypass the “Comments to the Author” section, enter your conflict of interest statement in the “Confidential to Editor” section, and submit your "Accept" recommendation.

Reviewer #2: All comments have been addressed

Reviewer #3: (No Response)

2. Is the manuscript technically sound, and do the data support the conclusions?

Reviewer #2: Yes

Reviewer #3: No

3. Has the statistical analysis been performed appropriately and rigorously? 

Reviewer #2: Yes

Reviewer #3: Yes

4. Have the authors made all data underlying the findings in their manuscript fully available?

Reviewer #2: Yes

Reviewer #3: No

5. Is the manuscript presented in an intelligible fashion and written in standard English?

Reviewer #2: Yes

Reviewer #3: Yes

6. Review Comments to the Author

Reviewer #2: The authors have made substantial efforts and included new data to answer questions and comments made by the reviewer.

Reviewer #3: Comments to the authors:

In this study, the authors evaluated the effects of nintedanib on gene expression in the lung of iRA-ILD model mice and revealed that Rfx2 and Bmpr2 were upregulated by nintedanib. But the authors could not confirm the RNA-seq data by RT-qPCR and western blotting using lung tissue.

Major comments

The authors revealed the effects of nintedanib on gene expression in the lung of iRA-ILD model mice and focused on two specific genes, Rfx2 and Bmpr2. They tried to confirm their RNA-seq data by RT-qPCR and western blotting using lung tissue, but there were no statistically significant changes by nintedanib. The authors mentioned nintedanib tended to alleviate the downregulation of Rfx2 and Bmpr2 gene expression or protein expression, but they did not show the p values. It is unclear how many lung tissues the authors used for the confirmation experiments but increasing the number of samples may make a significant difference.

RNA-seq results in a small number of cases often cannot be replicable. The author is also necessary to focus on other genes that popped up in RNA-seq data.

7. PLOS authors have the option to publish the peer review history of their article (what does this mean?). If published, this will include your full peer review and any attached files.

Reviewer #2: No

Reviewer #3: No

---

## [Author Response · Author response to Decision Letter 1]

31 May 2022

Responses to Reviewer 3

In this study, the authors evaluated the effects of nintedanib on gene expression in the lung of iRA-ILD model mice and revealed that Rfx2 and Bmpr2 were upregulated by nintedanib. But the authors could not confirm the RNA-seq data by RT-qPCR and western blotting using lung tissue.

Major comments

The authors revealed the effects of nintedanib on gene expression in the lung of iRA-ILD model mice and focused on two specific genes, Rfx2 and Bmpr2. They tried to confirm their RNA-seq data by RT-qPCR and western blotting using lung tissue, but there were no statistically significant changes by nintedanib. The authors mentioned nintedanib tended to alleviate the downregulation of Rfx2 and Bmpr2 gene expression or protein expression, but they did not show the p values. It is unclear how many lung tissues the authors used for the confirmation experiments but increasing the number of samples may make a significant difference.

(Response) We thank the reviewer for these insightful suggestions. We added p-values in Figures 4A and C, and Figures 5A and C. We have described the number of samples in the legends of Figures 4 and 5.

RNA-seq results in a small number of cases often cannot be replicable. The author is also necessary to focus on other genes that popped up in RNA-seq data.

(Response) We found 157 up- and downregulated genes by nintedanib treatment. We now evaluate expression of the several genes in fibroblasts. At the start of the study, we anticipated previously reported or downstream genes of the PDGFR, VEGFR, and FGFR signaling pathways; however, expression of these genes was not affected by nintedanib. In this study, we focused on a gene upregulated by nintedanib treatment, Rfx2, and showed that RFX2 was expressed mainly in the bronchial epithelium. In addition, a gene downregulated by nintedanib treatment, Muc5b, is shown to be expressed in the bronchioles and epithelial cells lining honeycomb cysts [references 44 and 45 in the manuscript]. We speculate that expression changes of these genes by nintedanib treatment may be associated with the differentiation of abnormal cell populations in the peripheral airways for alleviation of pulmonary fibrosis. Recently, Calabrese et al. reported RNA sequencing analysis of aggregates of fibroblasts lined by epithelial cells, which they called “epithelial cell (EC)/fibroblastic foci (FF) sandwich”, in lungs of patients with idiopathic pulmonary fibrosis and primary spontaneous pneumothorax as a control [reference 46]. They extracted EC/FF sandwiches using laser capture microdissection and performed RNA sequencing. They identified 23 upregulated genes in these lesions in idiopathic pulmonary fibrosis compared to those in lungs with primary spontaneous pneumothorax. Furthermore, they selected 14 upregulated genes with log fold-change ≥ 2 out of the 23 upregulated genes. Interestingly, five genes (Scgb3a1, Bpifb1, Pigr, Muc5B, and Krt5) of these 14 upregulated genes coincided with the downregulated genes by nintedanib treatment in the present study. Despite the differences in diseases, species, and sample preparation, these matches are intriguing and suggest that these genes could function in specific areas leading to the development of pulmonary fibrosis. Scgb3a1, Pigr, Bpifb1, and Muc5B are related to secretory and mucin proteins, and immune defenses [reference 46]. Scgb3a1 is shown to be expressed in secretory cells in idiopathic pulmonary fibrosis [reference 47]. The BPIFB1 protein was reported to be upregulated in the small airway epithelium in cystic fibrosis compared to that in the control [reference 48]. PIGR mediates immunoglobulin A for the immune exclusion of inhaled pathogens in bronchial mucosa [reference 49] We previously reported Krt5 expression in hyperplastic bronchiolar cells in a mouse model of bleomycin-induced interstitial pneumonia [reference 13]. Muc5b is expressed in the small airways and epithelial cells lining honeycomb cysts [references 44, 45]. We speculate that these findings suggest that these genes could be involved in the pathogenesis of pulmonary fibrosis or alleviation of pulmonary fibrosis by nintedanib treatment in specific lesions between the small airways and alveoli. Recent gene expression analyses in these lesions are noteworthy for unraveling the mechanism of the development of pulmonary fibrosis, and data accumulation is expected. Further investigations are needed to clarify the function of these five genes. 

We have made the following changes in the revised manuscript:

Discussion, page 22, line 337-340

“However, functional enrichment and pathway analysis showed that fibrosis-related genes such as Col1a1, Fn1, and Acta2 were not enriched in the terms "signaling pathway" and "focal adhesion," which are closely related to these tyrosine kinases,”.

Discussion, page 25-26, line 410-439

“Recently, Calabrese et al. identified 23 upregulated genes in aggregates of fibroblasts lined by epithelial cells, which they called epithelial cell/fibroblastic foci sandwich, in lungs with idiopathic pulmonary fibrosis compared to those in lungs with primary spontaneous pneumothorax by RNA sequencing analysis [46]. Intriguingly, five downregulated genes (Scgb3a1, Bpifb1, Pigr, Muc5B, and Krt5) by nintedanib treatment in the present study were included in the 14 upregulated genes with log fold-change ≥ 2 in the 23 upregulated genes. Scgb3a1, Pigr, Bpifb1, and Muc5B are related to secretory and mucin proteins, and immune defenses [46]. Scgb3a1 is expressed in secretory cells in idiopathic pulmonary fibrosis [47]. The BPIFB1 protein was reported to be upregulated in the small airway epithelium in cystic fibrosis compared to that in the control [48]. PIGR mediates immunoglobulin A for immune exclusion of inhaled pathogens in the bronchial mucosa [49]. We previously reported Krt5 expression in hyperplastic bronchiolar cells in a mouse model of bleomycin-induced interstitial pneumonia [13]. Muc5B is reported to be expressed in the small airways and epithelial cells lining honeycomb cysts [44, 45]. These findings imply that these genes could be involved in the development of pulmonary fibrosis and the effect of nintedanib in specific lesions between the peripheral airways and alveoli despite the differences in diseases, species, and sample preparation between the present study and Calabrese’s study. Further investigations are needed to clarify the functions of these genes.”

References

We added references 46-49.

To unify the term in the text,

Page 25, line 400

“Mucin5b” was changed to “Mucin5B (Muc5b)”.

Page 25, line 403

“Mucin5b” was changed to “Muc5b”.

To correct an error,

Page 26, line436

“iRA-ILS” was changed to “iRA-ILD”.

As requested by the journal office,

Materials and methods, Page 4-5, line 95-99

We added “All mouse experiments were performed according to the rules and regulations of the Fundamental Guidelines for Proper Conduct of Animal Experiments and Related Activities in Academic Research Institutions under the jurisdiction of the Ministry of Education, Culture, Sports, Science and Technology, Japan, and were approved by the Committee on the Ethics of Animal Experiments of Nagoya City University.”

---

## [Editor Report · Decision Letter 2]

3 Jun 2022

Nintedanib induces gene expression changes in the lung of induced-rheumatoid arthritis–associated interstitial lung disease mice

PONE-D-22-01668R2

Dear Dr. Uematsu,

We’re pleased to inform you that your manuscript has been judged scientifically suitable for publication and will be formally accepted for publication once it meets all outstanding technical requirements.

Kind regards,

Masataka Kuwana, MD, PhD

Academic Editor

PLOS ONE
---

## [Editor Report · Acceptance letter]

9 Jun 2022

PONE-D-22-01668R2 

Nintedanib induces gene expression changes in the lung of induced-rheumatoid arthritis–associated interstitial lung disease mice 

Dear Dr. Uematsu:

I'm pleased to inform you that your manuscript has been deemed suitable for publication in PLOS ONE. Congratulations! Your manuscript is now with our production department. 

Kind regards, 

on behalf of

Prof. Masataka Kuwana 

Academic Editor

PLOS ONE